# DyGRO-VLA: Cross-Task Scaling of Vision–Language–Action Models via Dynamic Grouped Residual Optimization

Sixu Lin [1 2]   Yunpeng Qing [3]   Litao Liu [4]   Ming Zhou [5]   Ruixing Jin [1]   Xiaoyi Fan [2]   Guiliang Liu [1 6 †]

## Abstract

Recent progress in Reinforcement Learning (RL) provides a principled approach to optimizing Vision-Language-Action (VLA) models, facilitating a shift from trajectory imitation to active learning in the task environment. Despite improvements in control precision, most RL optimizers remain task-specific, which reduces VLA models from generalist controllers to policies that overfit to a narrow set of tasks. In this study, we conduct an in-depth analysis of this phenomenon and highlight the importance of cross-task feature representations for improving the generalizability of VLA models. Motivated by this finding, we introduce DyGRO-VLA, a two-stage optimization framework that 1) effectively captures cross-task latent representations based on information-theoretic principles, and 2) dynamically refines policy optimization via a mixture-of-RL-residuals. DyGRO-VLA enables the RL optimizer to exploit task-relevant latent information while strategically mitigating adverse interference on the learned representations throughout the optimization process. We evaluate our approach on LIBERO, RoboTwin2 benchmarks, and further validate it on real world, demonstrating consistent improvements over strong baselines under multitask training and distribution shift. Our project page is available at `DyGRO-VLA`.

## 1. Introduction

Under the rapid development of large-scale foundation models, Vision-Language-Action (VLA) models have emerged as a promising paradigm for generalist robot manipulation policy by unifying perception, language understanding, and action control (Ma et al., 2024). Compared with lightweight task-specific policies (Chi et al., 2023; Zhao et al., 2023), VLAs typically offer stronger semantic grounding and better multi-task transfer, largely due to their higher representational capacity and the ability to reuse shared structure across heterogeneous tasks. As a result, VLAs are widely viewed as an important step toward general-purpose robots that can solve diverse real-world tasks with a unified model (Black et al., 2024; Intelligence et al., 2025).

Despite these advances, training a high-performing generalist policy solely from offline demonstrations remains challenging. Compared to datasets for language modeling and visual understanding, robotic datasets are relatively small, less diverse, and costly to collect (Xing et al., 2025), especially for long-tail scenarios that are less likely to be encountered in practice. To mitigate the data scarcity issues, recent studies incorporate online Reinforcement Learning (RL) to optimize VLA in the post-training (Chen et al., 2025b; Lu et al., 2025; Li et al., 2025a; Zang et al., 2025).

While RL-based optimization can effectively improve success rates on individual dexterous manipulation tasks (Luo et al., 2024; 2025), these gains often come at the expense of losing cross-task scalability. Our in-depth study reveals that as the number of tasks increases, the performance of RL optimization degrades substantially (see Figure 2). In some cases, it even harms the original VLA model that was pre-trained on offline data (see Figure 1). We find that this failure mode arises because the RL optimizer can distort the shared representation learned during pretraining, effectively isolating task-specific experience and reducing cross-task knowledge sharing (see Figure 3). This distortion leads to catastrophic forgetting and weakens cross-task adaptation. Therefore, balancing the trade-off between dexterity and generalizability remains a significant open challenge.

To address this challenge, we propose Dynamic Grouped Residual Optimization (DyGRO), a cross-task optimizer for VLA models. Specifically, DyGRO-VLA follows a two-stage optimization paradigm with the following objectives:

1) *Learning shared representations across tasks.* DyGRO-

---

[†]Corresponding author. [1]School of Data Science, The Chinese University of Hong Kong (Shenzhen) [2]Jiangxing Intelligence Technology Inc. [3]Zhejiang University [4]Rutgers University-New Brunswick [5]Shanghai AI Laboratory [6]Shenzhen Loop Area Institute. Correspondence to: Guiliang Liu < liuguiliang@cuhk.edu.cn>.

*Proceedings of the $43^{rd}$ International Conference on Machine Learning*, Seoul, South Korea. PMLR 306, 2026. Copyright 2026 by the author(s).

VLA follows the Information Bottleneck (IB) principle (Tishby et al., 1999) to extract action-relevant features from visual observations and robot states. By preserving useful information for guiding robot motion while discarding nuisance factors (e.g., background clutter and lighting variations), these features can be more effectively shared across tasks. Practically, to make IB learning computationally tractable, we derive a variational lower bound tailored to offline robot behavior modeling and use it to jointly train the feature representation and action prediction models.

2) *Finetuning with Mixture-of-RL-Residuals (MoRR).* To optimize the policy without disrupting the shared cross-task representation, we design the RL policy to predict a residual as refinements of the prediction from the action model (Johannink et al., 2019). However, classic residual learning is typically tailored to a single task. To generalize across a variety of tasks, we strategically combine residual refinements from a mixture of RL policies, where the mixture weights are determined by a dynamic routing mechanism. Consequently, while each policy only specializes in a subset of tasks, the router can adaptively select and compose them to produce the most effective refinement for the current tasks. To implement MoRR, we introduce a task-embedding method that provides task information to the router, based on which we design an online training scheme that jointly updates the RL residual policies and the routing model.

To validate the effectiveness of DyGRO-VLA, we conduct extensive experiments across diverse multi-task manipulation benchmarks, including LIBERO (Liu et al., 2023) and RoboTwin2 (Chen et al., 2025a). Across all settings, DyGRO-VLA consistently outperforms state-of-the-art baselines, achieving the highest average success rate of 97.1% on LIBERO, an absolute gain of +4.4% over its offline base model—and demonstrating substantial improvements on the most challenging LIBERO-Long suite (+9.8%). Furthermore, in real-world evaluations on RoboTwin2, DyGRO-VLA attains the best overall success in simulation (79.2%) and surpasses RFT baselines under Sim2Real transfer, particularly on complex bimanual and long-horizon tasks. These results highlight DyGRO-VLA as a scalable and robust solution for efficient multi-task reinforcement fine-tuning and real-world robotic manipulation.

## 2. Related Work

**Multi-task Learning.** A central challenge in multi-task learning is the interference among task gradients, which can hinder optimization efficiency. Prior work has sought to mitigate this issue through sample- or task-level reweighting strategies (Peng et al., 2023). Broadly, existing approaches fall into two categories: (1) designing reweighting schemes to balance the relative contributions of different tasks (Sener & Koltun, 2018; Parisotto et al., 2015; Yu et al., 2020), and

(2) adjusting per-task gradient directions to reduce conflicts and encourage cooperative learning (Yu et al., 2020; Liu et al., 2021; Chen et al., 2020; Wang et al., 2020; Bohn et al., 2024). More recently, token-level MoE routing methods have also been proposed to address gradient conflicts at finer granularity (Yang et al., 2024). (Yang et al., 2020; Huang et al., 2024; Kong et al., 2025; Wu et al., 2025) utilize routing strategy to resolve multi-task gradient conflicts.

**Reinforcement Learning for Vision-Language-Action Models.** Recent studies demonstrate that reinforcement finetuning can significantly improve VLA performance in manipulation tasks (Huang et al., 2025; Liu et al., 2025; Chen et al., 2025b; Guo et al., 2025; Tan et al., 2025; Li et al., 2025a; Zang et al., 2025). (Liu et al., 2025) analyzes the generalization benefits brought by RL to VLAs, while (Chen et al., 2025b) successfully deploys reinforced fine-tuning in real-world sparse-reward settings, showcasing effective adaptation in challenging robotic scenarios. In addition, (Guo et al., 2025) proposes an iterative learning paradigm that stabilizes training and improves efficiency. Different from these works, our method emphasizes the *scalability* of VLA post-training. While prior approaches focus on task-level performance gains or real-world adaptation, we argue that RL should preserve the intrinsic generalization ability of large VLAs, fundamentally distinguishing them from lightweight, task-specific models.

## 3. Problem Formulation

In this section, we discuss the problem of cross-task finetuning of VLA policies using RL methods and formulate the key challenges with empirical evidence.

**Multi-task Reinforcement Learning.** We consider solving a distribution of tasks $\zeta \sim p(\zeta)$ with a single policy $\pi$. In this setting, each task induces a POMDP $\mathcal{M}^\zeta = (\mathcal{S}^\zeta, \mathcal{A}, P^\zeta, R^\zeta, \mathcal{O}^\zeta, \rho^\zeta, \gamma)$. A task begins at an initial state $s_0 \sim \rho^\zeta$. At each timestep $t$, the agent receives an observation $o_t \in \mathcal{O}^\zeta$ composed of RGB images, robot proprioceptive information, and a natural language instruction. The agent then selects an action $a_t \in \mathcal{A}$ from the shared primitive action space, receives a reward $r_t \sim R^\zeta$, and transitions to the next state $s_{t+1}$ following the transition function $P^\zeta$. For brevity, we assume sparse rewards for robotic control: the agent receives a success reward of $+1$ upon task completion. The objective of the policy $\pi$ is to maximize the expected discounted return across the task distribution:

$$\pi^* = \arg\max_\pi \ \mathbb{E}_{\zeta \sim p(\zeta)} \ \mathbb{E}_{\tau \sim \pi, \mathcal{M}^\zeta} \Big[ \sum_{t=0}^{\infty} \gamma^t r_t \Big]. \quad (1)$$

Striving for temporal abstraction, we implement $\pi(\mathbf{a}_t \mid o_t)$ as *action chunking* policy (Zhao et al., 2023; Huang et al., 2025; Li et al., 2025b). An action chunk is defined as $\mathbf{a}_t = a_{t:t+h-1} \in \mathcal{A}^h$.

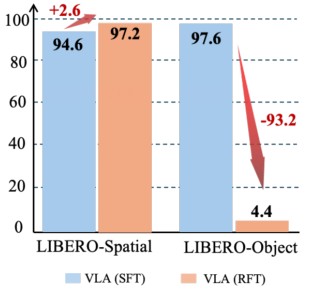

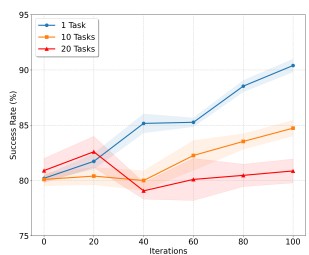

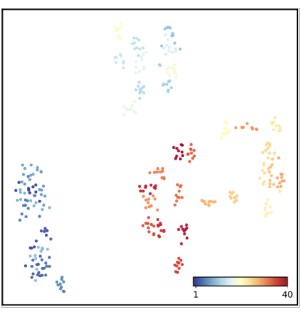

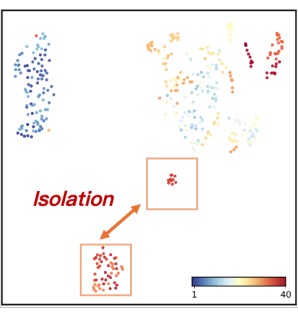

*(a)* Before RFT      *(b)* After RFT

*Figure 1*. Catastrophic Forgetting. RFT may improve the trained tasks but leads to increasing performance drops on other tasks.

*Figure 2*. Multi-Task Conflict. As the number of tasks increases, RFT becomes unstable and increasingly ineffective.

*Figure 3*. **t-SNE visualization of the shared feature space on LIBERO.** We embed representations from 40 tasks (10 samples per task) before and after single-task RFT.

**Catastrophic Forgetting in Cross-Tasks Adaptation.** Developing scalable and general-purpose VLA models is a central objective in embodied robotic control (Black et al., 2024; Shi et al., 2025; Intelligence et al., 2025). Although recent advances in reinforcement learning (RL) optimization have substantially improved the dexterity and task success rates of VLA policies (Li et al., 2025a; Lu et al., 2025), these models often remain highly task-specific. In particular, such RL optimizers tend to overfit to individual tasks and consequently struggle to retain prior knowledge in continual multi-task learning settings.

For empirical evidence, we test on LIBERO (Liu et al., 2023) benchmark. We conduct empirical analysis using the VLA model introduced in Section 4.1. As shown in Figure 1, when training a VLA on LIBERO-Spatial, we observe that performance on unrelated tasks in LIBERO-Object rapidly declines as training progresses. The success rate drops significantly, indicating that specialization on spatial tasks erodes the model's capacity to handle object-centric tasks.

Delving into the phenomenon of catastrophic forgetting, we observe that a fundamental prerequisite for the scalability of VLA models is the development of a shared feature space capable of encoding useful information across diverse tasks. However, during fine-tuning on a specific task, the RL agent often distorts this shared representation by isolating its representation of its experience from others. As shown in Figure 3, single-task RFT drives the tuned task's representations to form an isolated cluster, indicating a drift away from the shared feature space and reduced alignment with other similar tasks. It eventually results in a deterioration of VLA's cross-task competencies.

**Rethinking RL Post-training for VLA.** Revisiting the objective of VLA models, the ultimate goal is to learn a generalist control policy capable of performing effectively across diverse tasks. In practice, the variations among these tasks, including differences in time horizons, the number and types of target objects, and the visual, physical, and dynamic properties of the environments, are substantial (Kroemer et al., 2021; Kawaharazuka et al., 2025). Given the

inherently task-driven nature of RL optimizers, they are prone to overfitting to specific tasks and forgetting previously learned ones, as discussed above. This tendency naturally introduces a trade-off between achieving high task success rates and maintaining cross-task scalability. This trade-off poses a significant challenge for the direct application of RL optimizers in VLA models. To study this effect, we perform RL post-training with a classic actor–critic algorithm, Soft Actor-Critic (SAC) (Haarnoja et al., 2018). We evaluate three multi-task regimes (3 random seeds): (i) a single task from the LIBERO Spatial suite, (ii) 10 tasks from LIBERO Spatial, and (iii) 20 tasks from LIBERO Spatial and LIBERO Goal. As illustrated in Figure 2, the average success rate becomes increasingly unstable as the number of tasks grows during RL post-training. In particular, when trained on 10 tasks, only marginal improvements are observed, whereas with 20 tasks, the average success rate declines sharply, underscoring the limitations of current multi-task RL optimizers.

In this study, unlike prior works (Chen et al., 2025b; Lu et al., 2025; Li et al., 2025a) that focus on how RL optimizes the performance of individual tasks (i.e., for each task $\zeta$, $\max \mathbb{E}_{\mathcal{M}_\zeta}[\sum_{t=0}^{\infty} \gamma^t r_t]$), we examine the collective success rates of tasks as a group (i.e., for all $\zeta \sim p(\zeta)$, $\max \frac{1}{|\zeta|} \sum_\zeta \mathbb{E}_{\mathcal{M}_\zeta}[\sum_{t=0}^{\infty} \gamma^t r_t]$) and introduce a novel RL framework that effectively addresses the trade-off between accuracy and scalability in VLA post-training.

## 4. DyGRO-VLA

To achieve scalable cross-task performance, we propose DyGRO-VLA, a RL optimization framework for VLA models. Similar to recent VLA post-training methods (Lu et al., 2025; Li et al., 2025a;b), DyGRO-VLA follows an *offline pretraining*-to-*online fine-tuning* paradigm. However, unlike prior works targeting individual task performance, DyGRO-VLA seeks to generalize across multiple tasks by 1) learning shared representations in pretraining and 2) optimizing the performance all tasks concurrently during post-training.

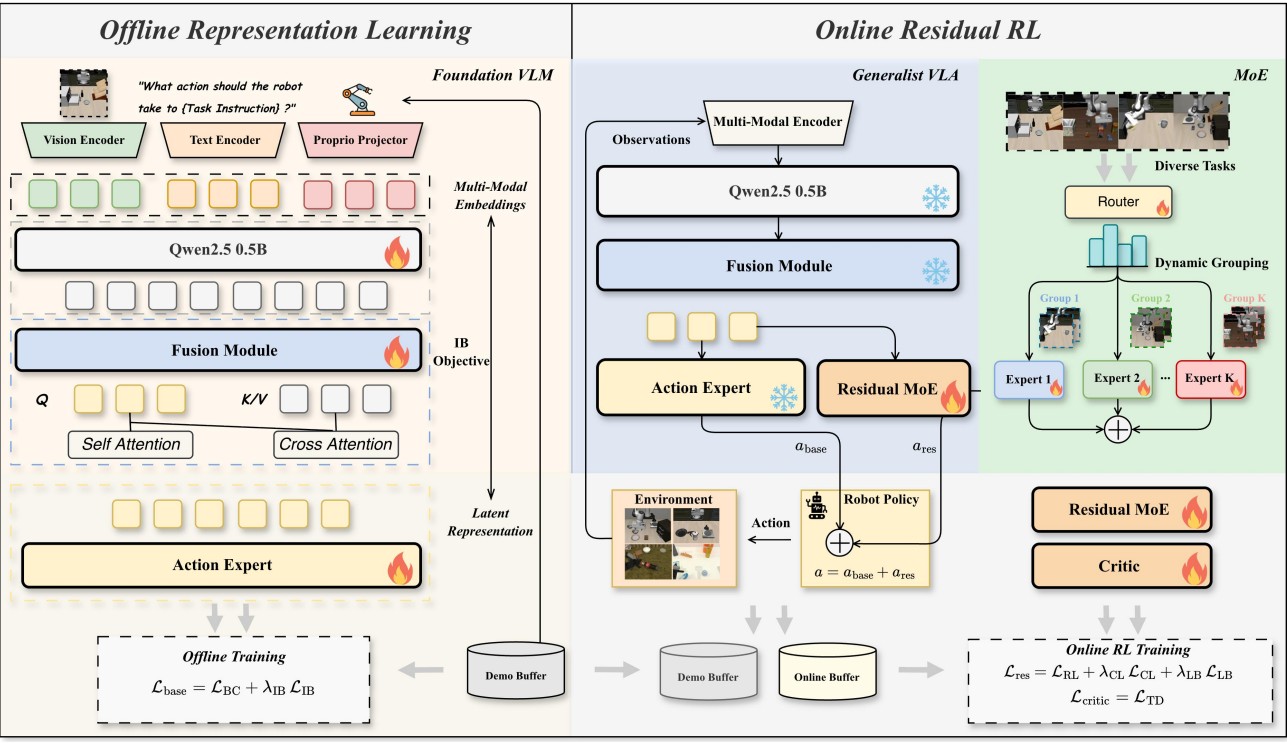

*Figure 4.* **Method pipeline.** DyGRO-VLA follows a two-stage training recipe. **1) Offline stage:** we train the VLA backbone to predict actions while learning a compact latent representation via an information-bottleneck (IB) objective. **2) Online stage:** we freeze the VLA backbone and optimize the residual MoE in online multi-task settings, serving as a residual compensation module on top of the base model to further improve overall multi-task ability and generalization.

## 4.1. Offline Pre-Training for Cross-Task Representation

The goal of pretraining is to learn a cross-task latent representation $p(z \mid o)$ that not only captures essential information from the observation $o$ but also facilitates the learning of the action model $\pi(a \mid z)$ across diverse tasks.

**Latent Feature Extraction.** DyGRO-VLA implements $p(z \mid o)$ using a VLM backbone coupled with a fusion module to encode exteroception from sensors and commands, and to fuse these features with proprioceptive information.

At each time step $t$, the backbone processes images from both wrist and third-view cameras $\mathcal{X}_t$ together with an instruction $\mathcal{L}_t$. Visual observations are encoded using DI-NOv2 (Oquab et al., 2024) and SigLIP (Zhai et al., 2023), and subsequently fused with tokenized language representations through a Qwen2.5-0.5B LLM (Team, 2024), yielding a sequence of hidden states. These hidden states are then partitioned into *conditioning tokens* $h_t^{\text{cond}}$ and *action tokens* $h_t^{\text{act}}$, where $h_t^{\text{cond}}$ *integrate* vision and language.

Upon completing the encoding stage, the fusion module initializes a set of learnable action-query tokens $q_t^a$ and fuses them with $h_t^{\text{cond}}$, $h_t^{\text{act}}$, and the proprioceptive input $p_t$ to produce $h_t^{\text{fuse}}$ (Wang et al., 2025). Appendix A.1 covers the details.

**Learning Task-Sharing Representation.** To effectively extract task-sharing features $\tilde{z}_T$ from the fused latent features $h_t^{\text{fuse}}$, DyGRO-VLA designs the learning objective based on the principle of Information Bottleneck (IB) (Tishby et al., 1999) and designs the learning objective as follows:

$$\max_{p_\theta(z|o)} I(Z; A) - \lambda_{\text{IB}} \, I(Z; O), \qquad (2)$$

where the probabilistic model $p_\theta(z|o)$ first extracts $h_t^{\text{fuse}}$ from $o$ and then transforms it into $\tilde{z}_T$. In essence, IB principle formalizes representation learning as a trade-off between *compression* and *task relevance*. Given an input observation $O$ and a latent representation $Z$, the IB framework seeks a stochastic encoder that retains information useful for downstream robot action prediction, such as spatial and morphological features of the target object, while discarding nuisance factors such as background and lighting conditions. This is achieved by maximizing $\mathcal{I}(Z; A)$ ($\mathcal{I}$ denotes the mutual information between two variables) while minimizing $\mathcal{I}(O; Z)$, weighted by $\lambda_{\text{IB}}$.

However, quantifying mutual information directly is intractable for high-dimensional inputs. Therefore, we employ a variational approximation to derive a lower bound of Objective 2 and propose the following objective:

**Proposition 4.1** (Variational Information Bottleneck for Ac-

tion Regression). *Let $Z$ be a latent representation produced by an encoder $p_\theta(z|o)$, and let $\pi_\theta(a|z)$ denote a probabilistic action decoder. Then, maximizing the IB objective (2) is equivalent to minimizing the following loss term:*

$$\mathcal{L}_{base} = \mathbb{E}_{p_\theta(z|o)}\big[-\log \pi_\theta(a|z)\big] + $$
$$\lambda_{\mathrm{IB}}\Big[\mathbb{E}_{P_{OZ}}[T_\psi(o,z)] - \log \mathbb{E}_{P_O P_Z}[e^{T_\psi(o,z)}]\Big] \quad (3)$$

*where 1) $T_\psi : \mathcal{O} \times \mathcal{Z} \to \mathbb{R}$ denotes a neural critic, and 2) $P_{OZ}$ and $P_O P_Z$ denote the joint distribution of O and Z, as well as the product of their marginals.*

Appendix B shows the proof. In practice, we estimate the $\mathbb{E}_{P_{OZ}}[T_\psi(o,z)]$ using paired samples in each mini-batch, and approximate sampling from the product of marginals $P_O P_Z$ by randomly permuting $z$ within the same mini-batch (pairing $o_i$ with $z_{\pi(i)}$ for a permutation $\pi$) (Belghazi et al., 2018). For stability, we reuse the same permutation within each optimization step.

Finally, to enable accurate action prediction, we parameterize the policy $\pi_\theta(a \mid z)$ using the action head architecture from prior work (Kim et al., 2025). Given the fused latent representation $z \in \mathcal{Z}$, the action head maps it to continuous control commands and outputs an action chunk of horizon $h$ for execution. Formally, the base policy is a mapping $\pi_\theta^{\mathrm{base}} : \mathcal{Z} \to \mathcal{A}^h$.

Accordingly, the maximum likelihood term in objective (3) can be approximated by regressing vector fields of fixed conditional probability path.

By combining $\pi_\theta^{\mathrm{base}}(\boldsymbol{a} \mid z)$ and $p_\theta(z \mid o)$, we obtain a shared base policy trained on offline demonstrations to acquire general multi-task competence.

## 4.2. Online Finetuning via Mixture of RL Residuals

Upon learning the task-sharing representation $p_\theta(z \mid o)$, we proceed to optimize the action prediction model $\pi_\theta^{\mathrm{base}}(\boldsymbol{a} \mid z)$ using an RL optimizer. However, as discussed in Section 3, although prior RL fine-tuning methods enhance the VLA's dexterity on individual tasks, these improvements come at the cost of distorting the learned task-sharing representation. Such distortion compromises the VLA model's ability to scale effectively across multiple tasks.

To address the trade-off between control dexterity and cross-task scalability, DyGRO-VLA employs a dynamic Mixture-of-RL-Residuals (MoRR) for multi-task RL optimization.

**Learning Task Embeddings.** DyGRO-VLA learns multiple RL policies and leverages their combination to optimize task performance. A key prerequisite for this process is determining which policies should be involved to handle a given task. To this end, we develop a task embedding denoted as $\tilde{z}_T$. Notably, unlike the latent representation

$p_\theta(z \mid o)$, which primarily captures decision-relevant patterns, the task embedding $\tilde{z}_T$ is designed to encode task-identification patterns. During training, task IDs are used only as supervision to shape the embedding space, while the router is conditioned on $\tilde{z}_T$ rather than directly consuming explicit environment labels (e.g., task IDs, names, or object categories). This design encourages the task embedding to capture semantic task features and allows DyGRO-VLA to select appropriate experts from the learned routing space at inference time.

Inspired by contrastive learning (Oord et al., 2018; Wu et al., 2025), we design the following loss function:

$$\mathcal{L}_{\mathrm{CL}}(\psi) = -\mathbb{E}_{(\tilde{z}_T, \zeta)}\left[\log \frac{\exp(\mathrm{sim}(\tilde{z}_T, e_\zeta)/\tau)}{\sum_{\zeta'=1}^N \exp(\mathrm{sim}(\tilde{z}_T, e_{\zeta'})/\tau)}\right],$$
$$(4)$$

where $N$ is the number of training tasks, $e_\zeta = \mathrm{Emb}(\zeta)$ denotes the learnable task prototype associated with task ID $\zeta$, $\tau$ is a temperature hyperparameter, and $\mathrm{sim}(\mathbf{a}, \mathbf{b}) = \mathbf{a}^\top \mathbf{b}$ is the similarity function. In practice, the MoE router is conditioned on the task embedding $\tilde{z}_T$, while the task prototypes $e_\zeta$ are only used during contrastive training to shape the embedding space and are not directly involved in routing decisions.

**Mixture of Residuals with Dynamic Routing.** Based on the task embeddings , DyGRO-VLA utilizes a Mixture-of-Experts (MoE) module (Jacobs et al., 1991; Shazeer et al., 2017) to refine the action chunk across multiple tasks.

Specifically, given a task embedding $\tilde{z}_T$, a router $h(\cdot)$ produces gating logits over $N$ residual experts $\{\pi_i^\Delta\}_{i=1}^N$ and activates only the top-$m$ of them. The corresponding gating weights are then used to combine the outputs of the selected experts as follows:

$$\Delta \boldsymbol{a} = \sum_{i=1}^m \omega_i(\tilde{z}_T)\,\tilde{\boldsymbol{a}}_i \ \ \text{s.t.} \ \ \tilde{\boldsymbol{a}}_i \sim \pi_{i,\psi}^\Delta(\tilde{\boldsymbol{a}}_i \mid z, \boldsymbol{a}_{\mathrm{base}}) \quad (5)$$

where 1) $\{\omega_i(\tilde{z}_T)\}_{i=1}^m$ are the top $m$ gating weights predicted by the router and 2) $\tilde{\boldsymbol{a}}_i$ is the residual chunk predicted by the $i$-th expert, conditioned on the latent features $z$ and the base action chunk $\boldsymbol{a}_{\mathrm{base}}$ (predicted the pre-trained policy $\pi_\theta^{\mathrm{base}}$). This residual parameterization preserves the base policy as a strong prior while enabling specialized experts to perform targeted corrections.

**Learning the Residual RL Policies.** Unlike prior residual-policy methods that attach an external residual controller to a frozen base policy (Xiao et al., 2025b; Yuan et al., 2024), we integrate a residual MoE into the base architecture by conditioning each expert on the same VLM latent representation used by the base action head. This avoids introducing a separate perception encoder for the residual branch, improving compute efficiency and reducing redundant re-encoding.

The RL loss of residual policy is formulated as:

$$\mathcal{L}_{\text{RL}}(\psi) = \mathbb{E}\left[\log \pi_{\psi,\theta}(\boldsymbol{a}_t \mid s_t) - \frac{1}{K}\sum_{k=1}^{K} Q_{\theta_k}(s_t, \boldsymbol{a}_t)\right]. \tag{6}$$

where 1) $\pi_{\psi,\theta}$ denotes the augmented policy that merges the predicted action refinement $\Delta \mathbf{a}$ (Eq. (5)) with action samples from the base policy $\pi_\theta^{\text{base}}(a \mid z)$ such that $\forall \boldsymbol{a} \sim \pi_{\psi,\theta}, \boldsymbol{a} = \Delta \boldsymbol{a} + \boldsymbol{a}_{\text{base}}$; and 2) $Q_{\theta_k}(s_t, \boldsymbol{a}_t)$ denotes the $k$-th critic in a $K$-ensemble of action-value functions, parameterized by $\theta_k$. The critic objective is detailed in Appendix A.2.

To mitigate routing instability and prevent expert collapse in the MoE policy, we regularize the gating distribution to encourage balanced expert utilization. Let $\hat{\omega}_n \in \Delta^{N-1}$ denote the predicted router distribution for the $n^{th}$ task embedding $\tilde{z}_{T,n}$ during training, and let $\hat{\omega}_{\text{avg}} = \frac{1}{B}\sum_{n=1}^{B} \hat{\omega}_n$ be the mini-batch average ($B$ is the minibatch size). We introduce an entropy-based load-balancing regularizer as:

$$\mathcal{L}_{\text{LB}}(\psi) = \sum_{i=1}^{N} \hat{\omega}_{i,\text{avg}} \log(\hat{\omega}_{i,\text{avg}} + \epsilon), \tag{7}$$

where $\epsilon$ is a small constant for numerical stability. The *sparse* mixture weights $\omega_i(\cdot)$ used in Eq. (5) are obtained by selecting top-$m$ $\hat{\omega}_i$ followed by renormalization. Minimizing $\mathcal{L}_{\text{LB}}$ is equivalent to maximizing the entropy of the routing probabilities, thereby preventing collapse to a single residual policy. We add $\mathcal{L}_{\text{LB}}$ to the policy objective. Therefore, the overall training objective of MoRR policy is:

$$\mathcal{L}_\pi(\psi) = \mathcal{L}_{\text{RL}}(\psi) + \lambda_{\text{CL}}\,\mathcal{L}_{\text{CL}}(\psi) + \lambda_{\text{LB}}\,\mathcal{L}_{\text{LB}}(\psi), \tag{8}$$

where $\lambda_{\text{CL}}$ and $\lambda_{\text{LB}}$ control the weights of the corresponding loss terms.

**Difficulty-Aware Sampling.** Inspired by (Lu et al., 2025; Florensa et al., 2017; 2018), we bias task sampling toward under-solved tasks. Specifically, given the empirical success rate $\text{sr}_j$ of task $\zeta_j$, we define $P(\zeta_j) \propto \exp\big(\max(0,\, target - \text{sr}_j)/\tau\big)$, which increases the sampling probability only when $\text{sr}_j < target$ and saturates otherwise. This asymmetric design focuses training on partially solvable yet still challenging tasks while retaining minimal exposure to mastered tasks.

**Practical Implementation.** In practice, we maintain two replay buffers: an offline buffer $\mathcal{D}_{\text{offline}}$ and an online buffer $\mathcal{D}_{\text{online}}$. Following RLPD-style sampling (Ball et al., 2023), we uniformly mix offline and online transitions for each gradient update. To ensure stable initialization of online residual RL, we warm up both the critic and policy prior to online interaction, as in Cal-QL (Nakamoto et al., 2023). We further use BC regularization (Fujimoto & Gu, 2021) and an advantage-filtered actor update (Nair et al., 2020) to stabilize residual policy learning. Pseudocode are provided in Appendix D.

## 5. Experiments

### 5.1. Experiments Settings.

**Benchmark.** 1) LIBERO (Liu et al., 2023) is a lifelong learning benchmark with five suites spanning 130 tasks. *LIBERO-Spatial* evaluates spatial generalization by varying object placements; *LIBERO-Object* tests object generalization by placing different objects into a box; *LIBERO-Goal* measures diverse operations in a fixed environment; and *LIBERO-Long* contains ten long-horizon tasks across varied scenes. 2) RoboTwin2.0 (Chen et al., 2025a) is a dual-arm manipulation benchmark designed for cross-embodiment evaluation. We select four representative tasks to assess real-world transfer performance and practical deployability.

**Baseline.** To assess each model's multi-task capability, we train all baseline methods on a unified training set formed by mixing data from all four suites with full shot data. To evaluate scalability and efficiency across model scales, we consider both generalist large VLA models and light-weight multi-task policies. Specifically, we include five recent large-scale VLA baselines reported in Table 1, including Octo (Team et al., 2024), OpenVLA (Kim et al., 2024), SpatialVLA (Qu et al., 2025), $\pi_0$-FAST* (Pertsch et al., 2025), and $\pi_0$ (Black et al., 2024). These models are fine-tuned from their released pretrained checkpoint when applicable. In addition, we evaluate two representative light-weight baselines, Diffusion Policy (Chi et al., 2023) and MT-ACT (Bharadhwaj et al., 2023). All baselines are trained under the same mixed-suite protocol for a fair comparison. To provide an initial sanity check for our approach, we also train the base DyGRO-VLA policy using supervised fine-tuning (SFT) under the same setting. Following common practice, we adopt LoRA (Hu et al., 2022) with rank 64 for this SFT baseline.

### 5.2. Main Results

**Results Analysis.** Table 1 reports success rates under the 4-suite co-training setting, evaluated on each LIBERO suite. Overall, DyGRO-VLA achieves leading or comparable performance across all suites. Compared with the offline SFT base model, our reinforcement fine-tuning consistently improves Spatial, Object, and Goal performance, and yields a clear average gain (92.7% → 97.1%). Notably, the largest improvement appears on the most challenging suite, *LIBERO-Long*, where success increases from 85.2% to 95.0%, indicating substantially enhanced long-horizon robustness. DyGRO-VLA remains competitive with prior multi-task VLA models under the same multi-task protocol.

**Reinforcement Fine-Tuning (RFT) Baseline Comparisons.** We evaluate DyGRO-VLA against representative RFT baselines on *LIBERO-Long* (Table 2) to characterize post-training performance. Since most prior RFT methods

*Table 1.* **LIBERO.** Success rate (%) on LIBERO Benchmark. **Best** results are in **bold** and **second-best** results are underlined.

| Method | Spatial | Object | Goal | Long | Avg. |
|---|---|---|---|---|---|
| **Multi-task SFT Models** | | | | | |
| Diffusion Policy (From Scratch) (Chi et al., 2023) | 59.6 | 73.8 | 51.6 | 41.0 | 56.5 |
| MT-ACT (From Scratch) (Bharadhwaj et al., 2023) | 50.2 | 72.0 | 60.2 | 50.2 | 58.2 |
| Octo (Team et al., 2024) | 78.7 | 85.5 | 84.2 | 51.0 | 74.9 |
| OpenVLA (Kim et al., 2024) | 82.1 | 87.3 | 77.4 | 51.8 | 74.7 |
| SpatialVLA (Qu et al., 2025) | 88.2 | 89.9 | 78.6 | 55.5 | 78.1 |
| $\pi_0$-FAST* (Pertsch et al., 2025) | 96.4 | 96.8 | 88.6 | 60.2 | 85.5 |
| $\pi_0$ (Black et al., 2024) | **98.0** | 96.8 | 94.4 | 88.4 | 94.4 |
| DyGRO-VLA (SFT) | 95.4 | 96.0 | 93.8 | 85.0 | 92.6 |
| **Multi-task RFT Models** | | | | | |
| DyGRO-VLA (Offline) | 95.6 | 96.0 | 94.0 | 85.2 | 92.7 |
| **DyGRO-VLA** | 97.6 | **98.6** | **97.2** | **95.0** | **97.1** |
| Δ | +2.0 | +2.6 | +3.2 | +9.8 | +4.4 |

*Table 2.* **RFT Comparisons.** Success rate (%) on LIBERO-Long.

| Method | LIBERO-Long |
|---|---|
| TGRPO (Chen et al., 2025c) | 59.2 |
| GRAPE (Zhang et al., 2024) | 57.2 |
| VLA-RL (Lu et al., 2025) | 59.8 |
| World-Env (Xiao et al., 2025a) | 57.8 |
| RIPT-VLA (Tan et al., 2025) | 93.8 |
| SimpleVLA-RL (Li et al., 2025a) | 91.7 |
| RLinf (Zang et al., 2025) | 94.0 |
| **DyGRO-VLA** | **95.2** |

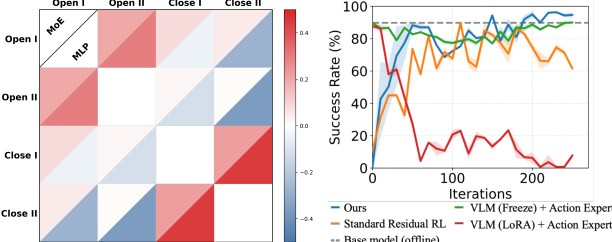

*Figure 5.* **Gradient Conflicts.** Pairwise cosine similarity between per-task gradients. Red indicates aligned gradients (synergy) and blue indicates conflicting gradients.

*Figure 6.* **Online Training Strategy Ablation.** Success rate versus online fine-tuning iterations for different strategies. Curves are averaged over 3 seeds; the dashed line denotes the offline base model.

are tailored to single-task or single-suite training and are not readily applicable to joint co-training across all four LIBERO suites, a direct four-suite comparison would be confounded by mismatched training assumptions. We therefore adopt a suite-wise evaluation protocol and report results consistently across suites. The results show DyGRO-VLA achieves the best performance with a success rate of 95.2%.

**Conflict Mitigation via Dynamic Grouping.** We analyze gradient conflicts using four representative tasks with contrasting interaction semantics, consisting of two *Open* tasks and two *Close* tasks (Open I/II, Close I/II). We quantify inter-task interference via pairwise cosine similarity of per-task gradients (Yu et al., 2020), comparing a single MLP residual head against our MoRR. As shown in Figure 5, gradients are positively aligned within the *Open* and *Close* pairs, but exhibit negative correlations across *Open↔Close*, indicating strong interference under a shared MLP head. MoRR reduces these negative similarities while maintaining within-group alignment, suggesting that MoE routing with dynamic grouping effectively mitigates cross-task conflicts.

### 5.3. Ablation Study

**Expert Number Ablation.** To verify whether scaling the MoRR improves multi-task performance, we vary the number of experts and report results in Table 4. Increasing

experts consistently boosts success: a single expert already reaches 93.4 average success, 4 experts improve to 97.0 with gains across all suites, and 8 experts achieve the best overall average performance (97.1 avg.), topping *Spatial* and *Goal* while tying the best *Long* result. Notably, the largest gain is on long-horizon tasks, suggesting that additional experts enable better specialization and mitigate cross-task interference.

**Component Ablation.** To assess the impact of each component in DyGRO-VLA, we perform a component-wise ablation (Tab. 5) by removing one term at a time while keeping all other settings fixed. Removing IB leads to a consistent drop in performance ($97.1 \rightarrow 96.8$), indicating that IB provides a reliable regularization benefit. Additionally, removing the InfoNCE-based contrastive learning objective leads to a pronounced degradation across suites, most notably on Long ($95.0 \rightarrow 90.4$), reducing the overall average to 94.3, which underscores the role of discriminative task embeddings for stable routing and multi-task generalization. Disabling difficulty-aware sampling also reduces performance ($97.1 \rightarrow 96.4$). Overall, the full model performs best, and these components contribute complementary gains.

*Table 3*. **RoboTwin2.** Success rate (%) on RoboTwin2 Benchmark. **Best** results are in **bold** and **second-best** results are underlined.

| Method | Beat Block Hammer | | Pick Dual Bottles | | Stack Bowls Two | | Place Empty Cup | | Avg. | |
|---|---|---|---|---|---|---|---|---|---|---|
| | Sim | Real | Sim | Real | Sim | Real | Sim | Real | Sim | Real |
| OpenVLA-oft (SFT) (Kim et al., 2025) | 54.0 | 15.0 | 32.0 | 25.0 | 88.0 | 70.0 | 50.0 | 60.0 | 56.0 | 42.5 |
| OpenVLA-oft (RFT) (Li et al., 2025a) | 71.9 | **30.0** | 54.0 | **40.0** | **92.0** | **90.0** | 96.1 | 60.0 | 78.5 | 55.0 |
| DyGRO-VLA | **72.2** | **30.0** | **57.0** | **40.0** | 90.4 | **90.0** | **97.0** | **70.0** | **79.2** | **57.5** |

*Table 4*. **Expert Number Ablation.** Success rate (%) on LIBERO.

| Method | Spatial | Object | Goal | Long | Avg. |
|---|---|---|---|---|---|
| DyGRO-VLA (1) | 95.2 | 96.4 | 93.6 | 88.2 | 93.4 |
| DyGRO-VLA (4) | 97.4 | **98.8** | 96.8 | 95.0 | 97.0 |
| DyGRO-VLA (8) | **97.6** | 98.6 | **97.2** | **95.0** | **97.1** |

*Table 5*. **Online Training Ablation.** Success rate (%) on LIBERO.

| Method | Spatial | Object | Goal | Long | Avg. |
|---|---|---|---|---|---|
| DyGRO-VLA | **97.6** | **98.6** | **97.2** | **95.0** | **97.1** |
| w/o IB Objective | 97.4 | 98.2 | 97.0 | 94.6 | 96.8 |
| w/o CL Objective | 95.2 | 97.2 | 94.2 | 90.4 | 94.3 |
| w/o DA Sampling | 97.4 | 98.4 | 97.0 | 93.0 | 96.4 |

**Online Training Strategy Ablation.** To examine the effect of online fine-tuning strategies, we conduct an ablation in which all methods are initialized from the same offline base model and evaluated over 3 seeds (Fig. 6). DyGRO-VLA achieves the highest success rate and the most stable improvement. In comparison, *Standard Residual RL* exhibits large oscillations and ends with noticeably lower performance. Freezing the VLM in *VLM (Freeze) + Action Expert* yields stable but modest gains, indicating limited adaptation capacity. Finally, *VLM (LoRA) + Action Expert* exhibits severe performance collapse, indicating that directly updating the backbone under large-scale multi-task online RL can substantially destabilize optimization.

## 5.4. Real-world Experiments

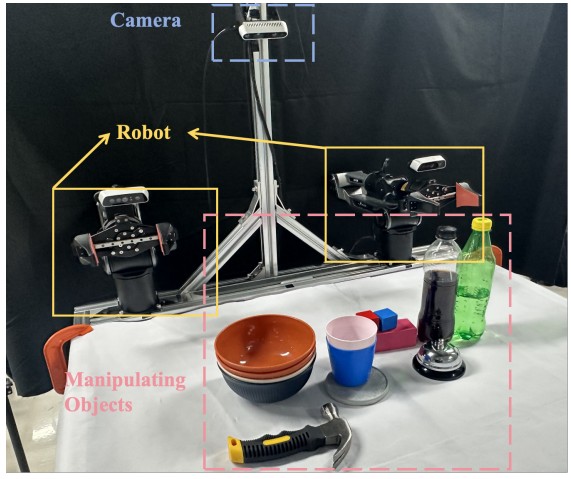

*Figure 7*. Real-world Platform for robotic manipulation.

**Real-World Settings.** We deploy our VLA model for real-world validation using a single Intel RealSense camera mounted in a head (top-down) view. DyGRO-VLA is trained in simulation and transferred to the real robot via a Sim2Real pipeline. Specifically, we follow the Sim2Real protocol of SimpleVLA-RL (Li et al., 2025a), applying domain randomization in simulation and transferring the resulting policy to the real world. We evaluate on four representative RoboTwin tasks: *Beat Block Hammer*, *Pick Dual Bottles*, *Stack Bowls Two*, and *Place Empty Cup*. We compare against OpenVLA-OFT (SFT) (Kim et al., 2025) and an RFT variant of OpenVLA-OFT trained with SimpleVLA-RL. All real-world evaluations use 20 trials per task, while simulation results follow the RoboTwin2 benchmark evaluation protocol.

**Results Analysis.** Our experiment results in Table 3 show that DyGRO-VLA achieves the best overall performance in simulation (79.2%) and matches or surpasses RFT baselines under Sim2Real transfer. In particular, DyGRO-VLA improves real-world success on *Pick Dual Bottles* and *Place Empty Cup*, while remaining competitive on *Stack Bowls Two*. These results validate the practical applicability of our method for deploying simulation-trained VLA policies in real-world settings.

## 5.5. Limitation

DyGRO-VLA has three main limitations: 1) *Online training sensitivity*: The online stage requires environment interaction and is sensitive to reward sparsity and hyperparameter tuning, which limits stability and sample efficiency. 2) *Task dependency in routing*: The routing mechanism relies on task embeddings trained with known task identities, reducing effectiveness in settings with ambiguous or unlabeled task boundaries. 3) *Limited real-world evaluation*: Current validation covers only a few tasks and a specific Sim2Real setup; broader testing across diverse tasks, embodiments, and sensing conditions is needed to assess generalization.

## 6. Conclusion

We investigate the scalability challenge of RL post-training for vision–language–action models, where multi-task online optimization can induce cross-task interference and catastrophic forgetting. To address this, we propose DyGRO-VLA, a two-stage framework that learns task-sharing representations from offline demonstrations and refines be-

havior online via dynamically routed residual RL experts. Across LIBERO, RoboTwin2, and real-world Sim2Real experiments, DyGRO-VLA improves multi-task performance and robustness over strong baselines, with notable gains on challenging tasks. These results highlight the value of preserving shared representations and using dynamic residual modularization for scalable VLA post-training. An important direction of future work is extending DyGRO-VLA to mobile manipulation tasks that involve locomotion tasks before manipulating objects.

## Impact Statement

This work aims to advance the field of machine learning and robotics by improving the efficiency and generalization of VLA models through Dynamic Grouped Residual Optimization (DyGRO). The proposed method could accelerate progress toward generalist robotic systems capable of performing a wider range of real-world tasks with greater adaptability and data efficiency.

Potential positive societal impacts include improved accessibility to automation, safer human-robot collaboration, and reduced data collection costs for robotic training. However, as with all advances in general-purpose AI and robotics, there are ethical considerations related to the potential misuse of autonomous systems, labor displacement in automation-sensitive industries, and biases originating from training data. We encourage future research to address these issues through transparent data practices, fairness-aware policy learning, and human-centered deployment standards.

## Acknowledgments

This work is supported in part by Shenzhen Science and Technology Program under grant KJZD20240903104008012, Shenzhen Science and Technology Program under grant ZDCY20250901113000001, CUHK-CUHK(SZ)-GDSTC Joint Collaboration Fund No. 2025A0505000053, and GuangDong Key Laboratory of Big Data Computing (2021B1212040002).

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

# A. DyGRO-VLA

## A.1. Fusion Module

Following prior work (Wang et al., 2025; Fu et al., 2025), each Fusion layer performs multi-head attention with the action query $q_t^a$ and keys/values from three sources: (1) $q_t^a$ itself, (2) the concatenation of the action hidden state $h_t^{\text{act}}$ and proprioceptive tokens $h_t^{\text{pro}} = \text{Enc}(p_t)$, and (3) the conditioning representation $h_t^{\text{cond}}$. A learnable scalar gate dynamically reweights the contribution of the conditioning branch, allowing the model to adaptively integrate task-relevant contextual information.

## A.2. Residual RL

**Residual MoE.** We parameterize the action distribution with an std model that predicts the standard deviation $\sigma_\theta$, while the Residual MoE provides the mean $\mu_\theta$. This formulation defines a factorized Gaussian policy $\pi_\theta(\mathbf{a} \mid s) = \mathcal{N}(\mu_\theta(s), \text{diag}(\sigma_\theta^2(s)))$, which enables stochastic actions during RL exploration and optimization.

**Residual Action Space.** When refining the base policy with a residual policy, we aim to keep the trajectory close to the original to avoid failure, which means the residual policy should make subtle adjustments. To control this, we bound the output within a specific range. We use the tanh function and mapping the residual action with a hyperparameter $\alpha$, ensuring it stays within $(-\alpha, \alpha)$.

**Offline-to-Online Critic Objective.** Our critic for residual RL is a lightweight ResMLP with attention pooling. Following Cal-QL (Nakamoto et al., 2023), we maintain an ensemble of $K$ critics $\{Q_{\theta_i}\}_{i=1}^K$ and optimize them with an $h$-step Bellman regression term together with a Cal-QL calibration regularizer:

$$\mathcal{L}_Q(\{\theta_i\}) = \frac{1}{K} \sum_{i=1}^K \Big( \mathcal{L}_{\text{TD}}(\theta_i) + \lambda_{\text{Cal}} \, \mathcal{L}_{\text{CalReg}}(\theta_i) \Big). \tag{9}$$

**Target ensemble and conservative aggregation.** For stable target estimation, we maintain a slowly-updated target ensemble $\{\bar{Q}_{\bar{\theta}_j}\}_{j=1}^K$ and define the aggregated target critic as

$$\bar{Q}_{\min}(s, \mathbf{a}) \;=\; \min_{j \in \{1, \ldots, K\}} \bar{Q}_{\bar{\theta}_j}(s, \mathbf{a}), \tag{10}$$

which yields the backup operator $\mathcal{B}^\pi \bar{Q}_{\min}$.

$h$-**step (chunk) Bellman regression.** Each critic is trained by regressing to the shared target:

$$\mathcal{L}_{\text{TD}}(\theta_i) = \mathbb{E}_{(s_t, \mathbf{a}_t, r_t^{(h)}, s_{t+h}) \sim \mathcal{D}} \Big[ \big( Q_{\theta_i}(s_t, \mathbf{a}_t) - \mathcal{B}^\pi \bar{Q}_{\min}(s_t, \mathbf{a}_t) \big)^2 \Big]. \tag{11}$$

The corresponding $h$-step Bellman backup is

$$\mathcal{B}^\pi \bar{Q}_{\min}(s_t, \mathbf{a}_t) = r_t^{(h)} + \gamma^h \, \mathbb{E}_{\mathbf{a}' \sim \pi(\cdot | s_{t+h})} \big[ \bar{Q}_{\min}(s_{t+h}, \mathbf{a}') \big], \tag{12}$$

where the $h$-step return is

$$r_t^{(h)} = \sum_{i=0}^{h-1} \gamma^i \, r_{t+i}. \tag{13}$$

**Cal-QL calibration regularizer.** To enable a smooth transition from offline data to online rollouts, we apply the Cal-QL calibration regularizer to each ensemble member, encouraging high value on policy actions while remaining anchored to dataset actions:

$$\mathcal{L}_{\text{CalReg}}(\theta_i) = \mathbb{E}_{s_t \sim \mathcal{D}} \Big[ \mathbb{E}_{\mathbf{a} \sim \pi(\cdot | s_t)} \big[ \max \big( Q_{\theta_i}(s_t, \mathbf{a}), V^\mu(s_t) \big) \big]$$
$$- \mathbb{E}_{\mathbf{a} \sim \mathcal{D}(\cdot | s_t)} \big[ Q_{\theta_i}(s_t, \mathbf{a}) \big] \Big]. \tag{14}$$

Here $\mathbf{a}_t = a_{t:t+h-1}$ denotes an action chunk and $\pi(\mathbf{a} \mid s)$ is the chunk-level policy. We approximate $V^\mu(s_t)$ using Monte Carlo returns from the offline dataset.

# B. Proof

**Proposition B.1** (Variational Information Bottleneck for Action Regression). *Let $Z$ be a latent representation produced by an encoder $p_\theta(z|o)$, and let $\pi_\theta(a|z)$ denote a probabilistic action decoder. Then, maximizing the IB objective (2) is equivalent to:*

$$\min_\theta \mathbb{E}_{p_\theta(z|o)}\Big[ - \log \pi_\theta(a|z)\Big] + \lambda_{\text{IB}}\Big[\mathbb{E}_{P_{OZ}}[T_\psi(o,z)] - \log \mathbb{E}_{P_O P_Z}[e^{T_\psi(o,z)}]\Big]$$

*where 1) $T_\psi : \mathcal{O} \times \mathcal{Z} \to \mathbb{R}$ denotes a neural critic, and 2) $P_{OZ}$ and $P_O P_Z$ denote the joint distribution of $O$ and $Z$, as well as the product of their marginals.*

*Proof.* Starting from the Information Bottleneck functional:

$$\max_{p_\theta(z|o)} I(Z;A) - \lambda_{\text{IB}} I(Z;O)$$

$$\overset{(1)}{=} \min_{p_\theta(z|o)} [-I(Z;A) + \lambda_{\text{IB}} I(Z;O)]$$

$$\overset{(2)}{=} \min_{p_\theta(z|o)} \mathbb{E}_{p(o,a)p_\theta(z|o)}[-\log p(a|z)] + \lambda_{\text{IB}} D_{\text{KL}}(P_{OZ}\|P_O P_Z)$$

$$\overset{(3)}{=} \min_{p_\theta(z|o)} \mathbb{E}_{(o,a)\sim\mathcal{D},\, p_\theta(z|o)}[-\log \pi_\theta(a|z)] + \lambda_{\text{IB}} \sup_{T_\psi} \left\{ \mathbb{E}_{P_{OZ}}[T_\psi(o,z)] - \log \mathbb{E}_{P_O P_Z}[e^{T_\psi(o,z)}]\right\}$$

$$\overset{(4)}{=} \min_{p_\theta(z|o)} \mathbb{E}_{(o,a)\sim\mathcal{D},\, p_\theta(z|o)}[-\log \pi_\theta(a|z)] + \lambda_{\text{IB}} \left( \mathbb{E}_{P_{OZ}}[T_\psi(o,z)] - \log \mathbb{E}_{P_O P_Z}[e^{T_\psi(o,z)}]\right) + \mathbb{C}$$

$$\overset{(5)}{=} \min_{p_\theta(z|o)} \mathbb{E}_{(o,a)\sim\mathcal{D},\, p_\theta(z|o)}[-\log \pi_\theta(a|z)] + \lambda_{\text{IB}} I_{\text{DV}}(O;Z) + \mathbb{C}$$

where $I_{\text{DV}}(O;Z) = \mathbb{E}_{P_{OZ}}[T_\psi(o,z)] - \log \mathbb{E}_{P_O P_Z}[e^{T_\psi(o,z)}]$ approximates $I(O;Z)$.

(1) holds because maximizing $I(Z;A) - \beta I(Z;O)$ is equivalent to minimizing its negative.

(2) holds by substituting the definitions $I(Z;A) = \mathbb{E}[\log p(a|z)] - \mathbb{E}[\log p(a)]$ and $I(Z;O) = D_{\text{KL}}(P_{OZ}\|P_O P_Z)$, where the constant $\mathbb{E}[\log p(a)]$ is omitted since it is independent of $\theta$.

(3) holds by applying the Donsker–Varadhan (DV) representation of the KL divergence:

$$D_{\text{KL}}(P_{OZ}\|P_O P_Z) = \sup_{T_\psi} \left\{ \mathbb{E}_{P_{OZ}}[T_\psi(o,z)] - \log \mathbb{E}_{P_O P_Z}[e^{T_\psi(o,z)}]\right\}.$$

This introduces the variational critic function $T_\psi(o,z)$.

(4) holds by absorbing the supremum into optimization over $\psi$ for notation consistency. The additive constant $\mathbb{C}$ encompasses all parameter-independent terms.

(5) holds by defining $I_{\text{DV}}(O;Z)$ as the Donsker–Varadhan estimate of mutual information. Thus, the resulting loss corresponds to the trainable Information Bottleneck formulation:

$$\boxed{\mathcal{L}_{\text{base}} = \mathbb{E}[-\log \pi_\theta(a|z)] + \lambda_{\text{IB}} \left( \mathbb{E}_{P_{OZ}}[T_\psi(o,z)] - \log \mathbb{E}_{P_O P_Z}[e^{T_\psi(o,z)}]\right).}$$

$\square$

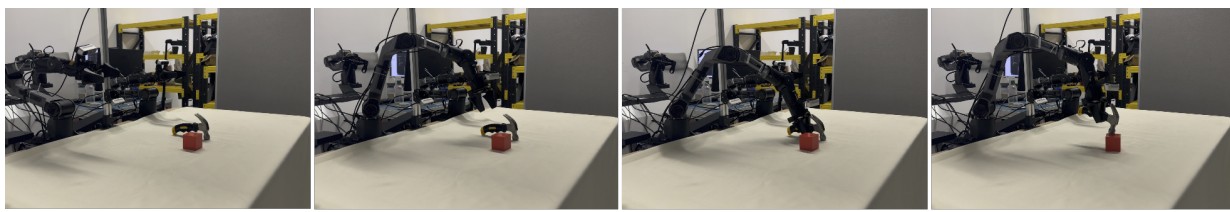

*(a)* Beat Block Hammer

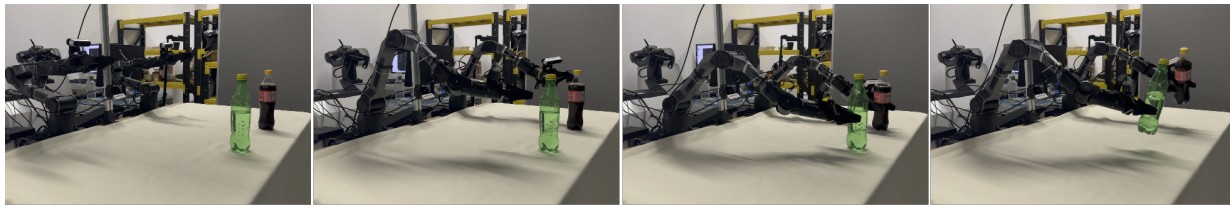

*(b)* Pick Dual Bottles

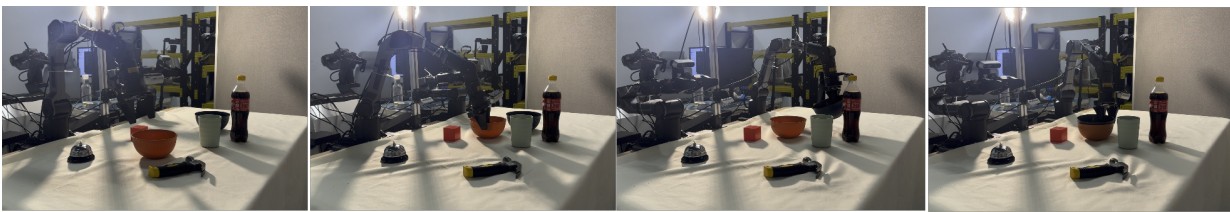

*(c)* Stack Bowls Two

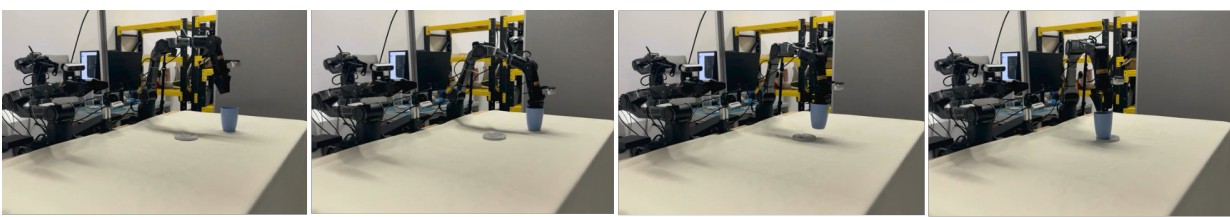

*(d)* Place Empty Cup

*Figure 8.* **Sim-to-real qualitative demonstrations of DyGRO-VLA on RoboTwin2.0.** The policy is trained in simulation and directly deployed in the real world. We show four real-world tasks: Beat Block Hammer, Pick Dual Bottles, Stack Bowls Two, and Place Empty Cup.

## C. Real-World Details

**Real-World Setups.** We deploy the training checkpoint zero-shot on the real robot without any real-world fine-tuning. We evaluate DyGRO-VLA on four real-world tasks from RoboTwin2.0: Beat Block Hammer, Pick Dual Bottles, Stack Bowls Two, and Place Empty Cup. These tasks cover diverse manipulation skills, including contact-rich tool-use, multi-object grasping, object stacking, and goal-conditioned placement. Beat Block Hammer requires accurate tool-object interaction, Pick Dual Bottles evaluates the ability to grasp multiple objects, Stack Bowls Two tests spatial alignment and stable placement, and Place Empty Cup examines pick-and-place generalization in real-world scenes.

**Real-World Visualization.** As shown in Figure 8, our proposed multi-task training framework achieves effective zero-shot sim-to-real transfer. After being trained only in simulation, DyGRO-VLA can be directly deployed on the real robot without additional real-world data or fine-tuning. The qualitative results demonstrate that the learned multi-task policy generalizes to diverse real-world manipulation scenarios, including tool-use, dual-object picking, stacking, and object placement.

## D. Pseudocode

The pseudocode of DyGRO-VLA is provided in Algorithm 1.

---

**Algorithm 1 DyGRO-VLA:** IB Pretraining + Online MoRR

---

**Require:** $\mathcal{D}_{\mathrm{demo}}, \mathcal{E}, \mathcal{T}; h, N_{\mathrm{on}}, U; \eta_{\mathrm{off}}, \eta_\pi, \eta_Q; \lambda_{\mathrm{IB}}, \lambda_{\mathrm{CL}}, \lambda_{\mathrm{LB}}; \theta, \nu, \phi, \psi$
**Ensure:** Executed chunk $a_t = a_t^{\mathrm{base}} + \Delta a_t$
 1: Initialize online replay buffer $\mathcal{D}_{\mathrm{on}} \leftarrow \emptyset$
 2: **Stage I: Offline pretraining**
 3: **for** $i = 1$ **to** $N_{\mathrm{off}}$ **do**
 4:     Sample $(o, l, p, a^\star) \sim \mathcal{D}_{\mathrm{demo}}$
 5:     Sample latent $z \sim p_\theta(z \mid o, l, p)$
 6:     Predict base chunk $a^{\mathrm{base}} \leftarrow \pi_\theta^{\mathrm{base}}(\cdot \mid z)$
 7:     $\mathcal{L}_{\mathrm{IB}} \leftarrow \text{DV-IB}(o, z; \nu)$                 // variational IB term
 8:     $\mathcal{L}_{\mathrm{base}} \leftarrow \mathcal{L}_{\mathrm{BC}}(a^{\mathrm{base}}, a^\star) + \lambda_{\mathrm{IB}}\,\mathcal{L}_{\mathrm{IB}}$
 9:     $(\theta, \nu) \leftarrow (\theta, \nu) - \eta_{\mathrm{off}}\,\nabla_{(\theta, \nu)}\mathcal{L}_{\mathrm{base}}$
10: **end for**
11: Freeze $\theta$                       // preserve task-sharing representation
12: **Stage II: Online finetuning**
13: **for** $k = 1$ **to** $N_{\mathrm{on}}$ **do**
14:     Reset $\mathcal{E}$ and sample task identity $\zeta \sim \mathcal{T}$
15:     **Rollout**
16:     **while not done do**
17:         Observe $(o_t, l_t, p_t)$
18:         $(z_t, \tilde{z}_t^T) \leftarrow \mathrm{Enc}_\theta(o_t, l_t, p_t)$           // latent + task embedding
19:         $a_t^{\mathrm{base}} \leftarrow \pi_\theta^{\mathrm{base}}(\cdot \mid z_t)$
20:         $\omega_t \leftarrow \text{Top-}m\text{Softmax}(\mathrm{Router}_\phi(\tilde{z}_t^T))$
21:         **for** each expert $i$ in top-$m(\omega_t)$ **do**
22:             Sample residual chunk $\tilde{a}_{t,i} \sim \pi_{i,\phi}^\Delta(\cdot \mid z_t, a_t^{\mathrm{base}})$
23:         **end for**
24:         $\Delta a_t \leftarrow \sum_{i \in \text{top-}m(\omega_t)} \omega_{t,i}\,\tilde{a}_{t,i}$
25:         Execute chunk $a_t = a_t^{\mathrm{base}} + \Delta a_t$ for $h$ steps, receive $(r_t^{(h)}, o_{t+h}, \texttt{done})$
26:         $(z_{t+h}, \tilde{z}_{t+h}^T) \leftarrow \mathrm{Enc}_\theta(o_{t+h}, l_{t+h}, p_{t+h})$
27:         Store $(z_t, \tilde{z}_t^T, \zeta, \omega_t, \Delta a_t, r_t^{(h)}, z_{t+h}, \texttt{done})$ into $\mathcal{D}_{\mathrm{on}}$
28:     **end while**
29:     **Updates**
30:     **for** $u = 1$ **to** $U$ **do**
31:         Sample minibatch $\mathcal{M} \sim \mathcal{D}_{\mathrm{on}}$
32:         $\mathcal{L}_Q \leftarrow \mathcal{L}_{\mathrm{TD/Cal\text{-}QL}}(\mathcal{M}; \psi, \theta, \phi)$
33:         $\psi \leftarrow \psi - \eta_Q\,\nabla_\psi \mathcal{L}_Q$
34:         $\mathcal{L}_{\mathrm{RL}} \leftarrow \mathcal{L}_{\mathrm{actor}}(\mathcal{M}; \psi, \theta, \phi)$       // SAC-style on augmented policy
35:         $\mathcal{L}_{\mathrm{CL}} \leftarrow \mathcal{L}_{\mathrm{InfoNCE}}(\mathcal{M}; \phi)$              // shape $\tilde{z}^T$
36:         $\mathcal{L}_{\mathrm{LB}} \leftarrow \mathcal{L}_{\mathrm{load\text{-}balance}}(\mathcal{M}; \phi)$        // prevent expert collapse
37:         $\phi \leftarrow \phi - \eta_\pi\,\nabla_\phi\Big(\mathcal{L}_{\mathrm{RL}} + \lambda_{\mathrm{CL}}\mathcal{L}_{\mathrm{CL}} + \lambda_{\mathrm{LB}}\mathcal{L}_{\mathrm{LB}}\Big)$
38:     **end for**
39: **end for**
40: **return** $\pi(a_t \mid o_t, l_t, p_t) = \pi_\theta^{\mathrm{base}}(\cdot \mid z_t) + \Delta a_t$

---

