# OpenReview forum: "DyGRO-VLA: Cross-Task Scaling of Vision–Language–Action Models via Dynamic Grouped Residual Optimization"
_ICML.cc/2026/Conference — ICML 2026 regular_

### Official Review · Reviewer_L9jp · 2026-03-08

**Soundness:** 3
**Presentation:** 3
**Significance:** 3
**Originality:** 3
**Overall Recommendation:** 5
**Confidence:** 3

**Summary:**

This paper proposed DyGRO-VLA and focused on cross-task performance of VLA models. The authors investigated the RL optimization performance under varied number of tasks and pointed out that knowledge isolation leads to "catastrophic forgetting", greatly reducing their performance. DyGRO-VLA addressed the issue by improving cross-task representation sharing and MoRR-based fine-tuning. The authors conducted cross-task experiments on different benchmarks and real-world tests; the results indicated good generalization and reliability in real-world. In general, I believe this model addresses an important issue with task transfer, enabling the model to utilize existing knowledge. The investigation of "catastrophic forgetting" is also valuable for the deployment of world & foundation models. Although there are minor issues that need to be addressed, I lean toward accept.

**Compliance With Llm Reviewing Policy:**

Affirmed.

**Key Questions For Authors:**

1. what happens when the environment changes but everything else stays the same?
2. does the ib latent actually remove factors such as lights? The authors did not provide related experiments or visual proof.
3. how robust is routing when instructions are unclear or the task is multi-stage?

**Limitations:**

yes

**Strengths And Weaknesses:**

Strength:
1. contrastive learning-inspired loss function is novel: by forming a task‑discriminative embedding space it stabilizes routing and reduces gradient conflict. It also seems to work well in mulit-task training, as reported by ablation studies.
2. the utilization of conditioning each expert on same VLM latents used by base action head reduces redundancy while retaining performance.
3. experiments section indicates the model yield good generalizability across task, mitigated the " Catastrophic Forget" situation.

Weakness:
1. the backbone uses Qwen-based models, which is not robotics‑specialized. There might be hallucinations or mis-grounding that cannot be addressed by IB objectives.
2. the design and underlying principles of task embedding zt~ is not clearly conveyed.

---

> ### Author Rebuttal · Authors · 2026-03-31
>
> Thanks for your thoughtful review and encouraging overall assessment. We especially appreciate your recognition of **the importance of the problem, our catastrophic-forgetting analysis, the novelty of the design, and the cross-task generalization results**. Below, we respond to each point.
>
> **W1. General-purpose Qwen backbone may have hallucination or mis-grounding issues that IB cannot fix**
>
> Thanks for raising this point. Hallucination or mis-grounding in the Qwen backbone is important, but orthogonal to the IB objective. Our focus is narrower: DyGRO improves **multi-task RL post-training** by reducing damage to shared multimodal representations. We therefore freeze the shared VLM backbone and place adaptation in lightweight routed residual branches rather than updating the full backbone (**Sec. 4.2**). This is consistent with **Fig. 6** and **Tab. 5**, where frozen-backbone residual adaptation is more stable than direct backbone adaptation. We have clarified that IB acts as a representation regularizer, not a complete solution to grounding failures of the underlying VLM.
>
> **W2. The design principle and role of the task embedding $z^T$ are not clearly conveyed**
>
> Thanks for raising this concern. In our design, $z^T$ is a **task-level embedding** for routing residual experts (**Sec. 4.3**). It is derived from the fused multimodal representation by pooling the latent sequence into a global task feature, followed by a lightweight projection head. During training, contrastive learning with task prototypes structures this space so that semantically similar tasks are grouped more coherently, helping the router assign experts more consistently, as reflected in **Fig. 5** and **Tab. 4**. We have clarified both the computation of $z^T$ and the train/test distinction: task identity is used only during training, while inference-time routing relies on the learned embedding itself.
>
> **Q1. What happens when the environment changes but everything else stays the same?**
>
> Thanks for this point. To address this, we add controlled evaluation on **LIBERO-Plus [1]**, using perturbations in **lighting**, **background texture**, and **object-related scene variation** while keeping the task and instruction unchanged. DyGRO achieves the best average performance and the strongest robustness under object-related perturbations, while remaining competitive under lighting and background changes.
>
> | Method | Light | Background | Objects | Average |
> |---|---:|---:|---:|---:|
> | pi0-fast | 73.2 | 73.2 | 68.8 | 71.7 |
> | pi0 | 85.0 | 81.4 | 68.9 | 78.4 |
> | DyGRO | **87.3** | 86.9 | **74.6** | **82.9** |
>
> **Q2. Does the IB latent actually remove factors such as lighting? No direct experiment or visual proof is provided**
>
> Thanks for this point. To provide direct evidence, we add a representation-level analysis. We keep the simulator state fixed, apply nuisance-only perturbations using the **lighting**, **background**, and **sensor noise** settings from LIBERO-Plus, and measure both the **mean cosine drift** of the learned latent (**lower is better**) and the **success rate**. We compare DyGRO-VLA with the variant without IB.
>
> As shown below, DyGRO-VLA consistently achieves both **smaller latent drift** and **higher success rate** under these nuisance perturbations. This provides direct evidence for improved **invariance to nuisance perturbations**.
> | Method | Light Succ. ↑ | Light Drift ↓ | Background Succ. ↑ | Background Drift ↓ | Noise Succ. ↑ | Noise Drift ↓ |
> |---|---:|---:|---:|---:|---:|---:|
> | DyGRO-VLA w/o IB | 85.0 | 0.0112 | 84.3 | 0.0214 | 54.2 | 0.0394 |
> | DyGRO-VLA | **87.3** | **0.0108** | **86.9** | **0.0109** | **57.0** | **0.0257** |
>
> **Q3. How robust is routing when instructions are unclear or the task is multi-stage?**
>
> Thanks for this concern. We treat these as two separate cases.
>
> For **multi-stage tasks**, the strongest current evidence is **LIBERO-Long**, where DyGRO achieves the largest gain among all suites (**Tab. 1**) and also outperforms representative RFT baselines (**Tab. 2**). This suggests that routing is helpful in more temporally extended and compositional settings, although this remains indirect evidence.
>
> For **unclear instructions**, we use the **Language Instruction** perturbation in **LIBERO-Plus [1]** as a partial proxy for robustness to **language-side variation**. This perturbation rewrites the original instruction while preserving task semantics, including contextual distraction, commonsense-based re-description, and reasoning-chain variation. Under this setting, DyGRO achieves the best performance in both the original (**97.1**) and perturbed (**61.4**) settings, suggesting robustness to semantically preserved instruction variation.
> | Method | Original | Language |
> |---|---:|---:|
> | pi0 | 94.4 | 58.8 |
> | pi0-fast | 85.0 | 61.0 |
> | DyGRO | **97.1** | **61.4** |
>
> **Reference:**
>
> [1] Fei et al. *LIBERO-Plus: In-depth robustness analysis of vision-language-action models*.

---

> > ### Author Rebuttal · Reviewer_L9jp · 2026-04-03
> >
> > My concerns are fully resolved.

---

> > > ### Author Response · Authors · 2026-04-03
> > >
> > > Thank you for your positive feedback. We are glad that our response addressed your concerns, and we sincerely appreciate your recognition of our work.

---

### Official Review · Reviewer_B9pZ · 2026-03-11

**Soundness:** 2
**Presentation:** 2
**Significance:** 2
**Originality:** 2
**Overall Recommendation:** 3
**Confidence:** 4

**Summary:**

- This paper identifies a key failure mode in RL post-training for VLA models: as the number of training tasks increases, multi-task RL becomes unstable and can damage cross-task generalization by distorting shared representations.
- It proposes DyGRO-VLA, a two-stage approach: (1) an offline Information-Bottleneck-style latent representation to preserve action-relevant information, and (2) an online RL phase that freezes the shared representation and improves behavior via a mixture of residual RL experts with dynamic routing to reduce cross-task interference.
- Experiments on LIBERO / LIBERO-Long (and a limited Sim2Real setting) suggest the method improves multi-task and long-horizon performance and reduces negative task-to-task gradient interference compared to shared residual fine-tuning

**Compliance With Llm Reviewing Policy:**

Affirmed.

**Final Justification:**

The rebuttal has addressed my concerns.
I misunderstood that they were trying to do real-world RL fine-tuning, but I realized they post-train VLA via RL in simulation and transfer to the real-world, which does not need to consider real-world RL cost.
I'll raise my score toward the positive side.

**Key Questions For Authors:**

Questions are mostly aligned with weaknesses.

- **Real-world RL cost vs alternatives:** What are the concrete real-world costs of the online RL stage (wall-clock time, number of online episodes/rollouts, resets, human intervention) and how do these compare to a baseline that collects a few additional demonstrations and runs SFT? Under what conditions does RL provide a clear advantage over “few-demo SFT” in terms of time-to-performance?
- **Motivation for RL on “easy-to-demo” tasks:** For tasks that are straightforward to demonstrate, what is the motivation to use online RL rather than simply augmenting demonstrations and re-running SFT (or preference-based fine-tuning)? Are there task categories where RL is consistently necessary in your setting?
- **Effect size and practical significance:** Can you report per-suite/per-task absolute improvements, confidence intervals across seeds, and the proportion of tasks where the method yields a meaningful gain (not just a marginal increase in already-high success rates)?
- **Fairness and generalization:** Since baseline VLAs benefit from large-scale diverse pretraining, while your approach appears to lean on in-domain demonstrations plus in-domain online interaction, can you add evaluations that explicitly test generalization beyond the tuning distribution (unseen tasks, new objects, novel instructions, different environments/embodiments)? Even if matching baseline pretraining data is infeasible, a rigorous generalization comparison would help interpret the results.
- **Overfitting diagnostics:** Do you observe signs of domain overfitting (e.g., performance drop on held-out suites/tasks or shifts) when increasing online RL steps? If so, what mechanisms prevent over-specialization, and can you provide curves showing in-domain gains vs out-of-domain degradation?

**Limitations:**

- Requires online RL interaction and is sensitive to reward sparsity and hyperparameter tuning, which can limit stability and sample efficiency.
- Routing may rely on assumptions about task structure/identity, reducing robustness in settings with ambiguous or unlabeled tasks.
- Real-world evidence is limited in breadth; generalization across robots, sensors, and environmental variation remains uncertain.

**Strengths And Weaknesses:**

## Strengths

- Articulates and empirically demonstrates a practical issue: multi-task RL post-training can degrade as tasks scale, including representation drift and forgetting.
- The proposed solution is well-aligned with the diagnosis: freeze shared representations + residual RL, reducing the risk of corrupting pretrained knowledge.
- Mixture-of-residual-experts with routing is a reasonable mechanism to mitigate gradient conflicts and cross-task interference.

## Weaknesses

- **Real-world online RL practicality is under-discussed:** Online RL on physical robots is typically expensive in time, wear-and-tear, and human supervision, yet the paper provides limited discussion of this trade-off. It is unclear whether the proposed online RL stage is justified for tasks where a small number of additional demonstrations with SFT could likely achieve similar gains much faster.
- **Lack of concrete “real-world RL cost” accounting:** The paper does not provide a clear breakdown of real-world requirements (e.g., wall-clock hours, number of online episodes/rollouts, intervention rate, reset time), nor does it compare these costs against a “few-demo SFT” alternative under comparable performance targets.
- **Unclear magnitude/meaningfulness of gains vs baselines:** While improvements are reported, it is not always obvious whether the gains are consistently large and practically significant across suites/tasks (especially given near-saturation regimes where absolute improvements can be small).
- **Ablations appear to show modest deltas:** Several ablation variants seem to yield relatively small performance differences, raising questions about whether each proposed component (e.g., IB objective, routing losses, load-balancing, difficulty-aware sampling) is truly critical or whether the method is over-parameterized relative to the observed marginal benefits.
- **Potential fairness concern in pretraining vs tuning setup:** Strong VLA baselines are typically pretrained on large, diverse datasets and then lightly tuned with small in-domain data, whereas the proposed approach appears to rely more heavily on in-domain demonstrations and in-domain online interaction. This could make the method more prone to domain overfitting and complicate fairness when the goal is to claim improved general cross-task generalization.
- **Generalization evaluation is limited relative to the claims:** Given the emphasis on cross-task scaling and preserving generality, more explicit unseen-task / distribution-shift evaluations (e.g., new objects, instructions, environments, embodiments) would strengthen the argument that gains are not primarily in-domain overfitting.

---

> ### Author Rebuttal · Authors · 2026-03-30
>
> Thank you for the insightful feedback. We respond point by point below.
>
> **W1/W2/Q1. Real-world RL cost and comparison to few-demo SFT**
>
> Thanks for this concern. Our method does **not** perform on-robot RL. Online RL is conducted entirely in simulation; the real-world section is a **Sim2Real evaluation** following SimpleVLA-RL. Therefore, on-robot RL wall-clock time, real-world rollouts, resets, and intervention during training are not applicable here. More importantly, we do not claim that real-world online RL is broadly practical or preferable to collecting more demonstrations and re-running SFT. Rather, our focus is how to preserve shared VLA representations and scale across many tasks under RL post-training. We have revised Sec. 5.4 to clarify this scope.
>
> **W3/Q3. Effect size and practical significance**
>
> Thanks for this concern. As reported in **Table 1**, our gains are most evident in the more challenging **multi-task** setting. On the 4-suite co-training benchmark, DyGRO-VLA improves the average success rate from **92.7** to **97.1** (**+4.4**), with consistent gains across all four suites: **Spatial +2.0**, **Object +2.6**, **Goal +3.2**, and **Long +9.8**. In particular, on **LIBERO-Long**, success increases from **85.2** to **95.0**, reducing the failure rate from **14.8** to **5.0** (about **66% relative reduction**). On the full 4-suite average, the failure rate is reduced from **7.3** to **2.9** (about **60% relative reduction**). In addition, **Fig. 6** shows improved stability across **3 seeds**.
>
> **W4. Ablation strength and whether all components are critical**
>
> Thanks for this point. The **IB objective** and **difficulty-aware sampling** mainly serve as **auxiliary stabilizers**, so their standalone deltas are naturally smaller.
>
> They address different failure modes: **IB** improves representation robustness, while **difficulty-aware sampling** improves optimization stability. Their role is therefore complementary rather than dominant. To make this more concrete, we additionally evaluate the effect of the **IB** term under nuisance-only perturbations from **LIBERO-Plus**. The corresponding results are provided in **Table A** at our anonymous website (https://anonymous.4open.science/w/DyGRO/). We have revised the discussion accordingly to reflect this hierarchy more clearly.
>
> **W5/W6/Q4. Fairness and generalization beyond the tuning distribution**
>
> Thanks for this concern. Our comparison is primarily at the **downstream post-training level**, not a claim that our foundation model is intrinsically stronger. The strongest evidence is the gain from **DyGRO-VLA (Offline)** to **DyGRO-VLA** under the same setup, plus improvements over representative RFT baselines. The current submission also supports **cross-suite scaling** and **Sim2Real transfer** more directly than broad open-world generalization.
>
> To further address this point, we have added OOD evaluations on **LIBERO-Plus [1]** across four perturbation settings (**Light, Background, Objects,** and **Language**), where DyGRO-VLA also demonstrates stronger robustness under domain shifts.
> | Method | Light | Background | Objects | Language | Average |
> |---|---:|---:|---:|---:|---:|
> | pi0-fast | 73.2 | 73.2 | 68.8 | 61.0 | 69.1 |
> | pi0 | 85.0 | 81.4 | 68.9 | 58.8 | 73.5 |
> | DyGRO | **87.3** | **86.9** | **74.6** | **61.4** | **77.6** |
>
> **Q2. Motivation for RL on “easy-to-demo” tasks and few-demo SFT**
>
> Thanks for this important point. We do **not** intentionally choose “easy-to-demo” tasks. Instead, we use **LIBERO** and **RoboTwin2** because they are **standard benchmarks for VLA post-training**, enabling controlled and broadly comparable evaluation of post-training methods. Our goal is therefore to study a different question: **under a standard RL-based VLA post-training protocol, how can post-training scale across tasks without harming the shared generalist representation?**
>
> We have revised the paper to clarify this scope.
>
> **Q5. Overfitting diagnostics**
>
> Thanks for this point. Standard RL Finetung (RFT) often overfits to the learning environment. DyGRO mitigates this through a **frozen shared backbone**, **residual adaptation**, and **dynamic routing**. The frozen backbone preserves the shared multimodal representation, residual adaptation limits online updates to small corrective refinements, and dynamic routing reduces cross-task interference across experts. This helps improve in-domain performance while alleviating catastrophic forgetting on other domains.
>
> To make this more explicit, we have added curves comparing standard RFT and DyGRO-VLA, where both methods are post-trained on **LIBERO-Spatial** and evaluated on **LIBERO-Spatial** and **LIBERO-Object**. The corresponding figure is provided as **Fig. A** on our anonymous website.
>
> **References:**
>
> [1] Fei S, Wang S, Shi J, et al. LIBERO-Plus: In-depth robustness analysis of vision-language-action models.

---

> > ### Author Rebuttal · Reviewer_B9pZ · 2026-04-05
> >
> > Thank you for the clarifications. However, my main question remains about the motivation for RL post-training itself.
> >
> > More specifically, while I understand the authors’ goal is to preserve shared VLA representations under RL post-training, it is still unclear why RL post-training is the necessary or preferred adaptation mechanism in the first place, compared with other fine-tuning strategies such as collecting a modest number of additional demonstrations and continuing SFT. RL may indeed help cover behaviors or corrective strategies not present in the demonstrations, but that advantage comes together with well-known drawbacks that I mentioned in the questions.
> >
> > Without a clearer discussion of why these trade-offs are worthwhile here, the motivation for introducing RL post-training remains under-justified. I would therefore encourage the authors to more explicitly explain in what regime RL post-training provides unique value over simpler alternatives, and why that value is compelling enough to warrant the additional complexity.

---

> > > ### Author Response · Authors · 2026-04-07
> > >
> > > Thank you for this important follow-up. In the setting we study, RL post-training is particularly worthwhile when a strong offline policy is already available, but further gains require handling **recovery, execution drift, long-horizon credit assignment, and robustness to distribution shift**—regimes that are difficult to cover efficiently with a modest amount of additional demonstrations. To explicitly clarify this point, we additionally include a **continued-SFT baseline** with a modest extra demonstration budget (**+20 demos per task**). We use this modest budget to directly reflect the practical regime raised by the reviewer, namely whether a small amount of extra imitation data can serve as a simpler alternative to RL post-training.
> > >
> > > Starting from the same offline base, continued SFT improves the average LIBERO success rate only slightly from **92.6** to **92.9**, and improves **LIBERO-Long** from **85.0** to **85.4**. In contrast, RL post-training improves the same base to **97.1** average success, with the largest gain on **LIBERO-Long** (**85.0 -> 95.0**), as shown in **Table A**. This comparison suggests that an important benefit of RL comes from enabling further improvement through online interaction beyond the limited coverage of a modest number of additional demonstrations.
> > >
> > > **Table A. Comparison with additional demonstrations**
> > > | Method | Spatial | Object | Goal | Long | Avg. |
> > > |---|---:|---:|---:|---:|---:|
> > > | DyGRO-VLA (SFT) | 95.4 | 96.0 | 93.8 | 85.0 | 92.6 |
> > > | DyGRO-VLA (Continued SFT) | 95.6 | 96.2 | 94.2 | 85.4 | 92.9 |
> > > | DyGRO-VLA | **97.6** | **98.6** | **97.2** | **95.0** | **97.1** |
> > >
> > > We believe this difference arises because **continued SFT primarily improves behaviors that are explicitly covered by the added trajectories**, whereas a modest number of demonstrations may still have limited coverage of **newly encountered failure states, long-tail intermediate states, and recovery situations induced by execution drift**. By contrast, RL can continue improving from task-level reward feedback and discover corrective strategies through interaction. This intuition is also consistent with recent VLA post-training results. For example, *iRe-VLA* [1] explicitly studies how to improve pretrained VLA models through online RL; *SimpleVLA-RL* [2] improves the average LIBERO success rate of SFT-tuned OpenVLA-OFT from **91.0** to **99.1**; and *RIPT-VLA* [3] shows that even from a very weak one-demo initialization, task success can be rapidly improved using only sparse binary rewards. These prior results, together with our added continued-SFT baseline, support the view that RL is a practical post-training mechanism once a strong offline base is already available.
> > >
> > > Another important benefit of RL is its improved **robustness to OOD scenarios**. Prior work [4] suggests that RL fine-tuning can improve generalization, especially in terms of semantic generalization and execution robustness. Motivated by this, we additionally evaluate on **LIBERO-Plus** under OOD perturbations such as lighting, background, and sensor-noise shifts. As shown in **Table B**, DyGRO-VLA improves the OOD average from **68.7** to **77.1**, with gains across all three perturbation types, while continued SFT with extra demonstrations brings only negligible OOD improvement (**68.7 -> 68.8**). These results further suggest that the value of RL post-training is not limited to fitting more trajectories, but also lies in improving robustness beyond what can be efficiently covered by an additional demonstration budget.
> > >
> > > **Table B. OOD robustness under LIBERO-Plus perturbations**
> > >
> > > | Method | ID Avg. | Light | Background | Noise | OOD Avg. |
> > > |---|---:|---:|---:|---:|---:|
> > > | DyGRO-VLA (SFT) | 92.6 | 80.4 | 77.2 | 48.6 | 68.7 |
> > > | DyGRO-VLA (Continued SFT) | 92.9 | 80.6 | 77.0 | 48.8 | 68.8 |
> > > | DyGRO-VLA | **97.1** | **87.3** | **86.9** | **57.0** | **77.1** |
> > >
> > > Therefore, our point is **not** that RL universally dominates simpler alternatives. Rather, in the VLA post-training setting we study, RL provides a practical way to further improve a strong offline base, particularly for adapting to failure states, recovery behaviors, long-horizon tasks, and distribution shift. Our contribution is to make such RL adaptation effective for generalist VLAs while preserving shared cross-task generalization. We thank the reviewer again for raising this important point and hope that these clarifications help address the concerns.
> > >
> > > **References**
> > >
> > > [1] Guo, Y., Zhang, J., Chen, X., et al. *Improving Vision-Language-Action Model with Online Reinforcement Learning*. ICRA, 2025.
> > > [2] Li, H., Zuo, Y., Yu, J., et al. *SimpleVLA-RL: Scaling VLA Training via Reinforcement Learning*. ICLR, 2026.
> > > [3] Tan, S., Dou, K., Zhao, Y., and Krähenbühl, P. *Interactive Post-Training for Vision-Language-Action Models*. arXiv, 2025.
> > > [4] Liu, J., Gao, F., Wei, B., et al. *What Can RL Bring to VLA Generalization? An Empirical Study*. NeurIPS, 2025.

---

### Official Review · Reviewer_dNXG · 2026-03-13

**Soundness:** 3
**Presentation:** 3
**Significance:** 2
**Originality:** 3
**Overall Recommendation:** 4
**Confidence:** 4

**Summary:**

The paper's broad aspect pertains to improving cross-task scalability of RL post-training for VLA policies, motivated by observed catastrophic forgetting and instability as the number of tasks increases. The work intends to present a central area solution: a two-stage recipe that learns a shared representation with an Information Bottleneck style objective during offline behavior modeling, then freezes the backbone and performs online RL through a Mixture-of-RL-Residuals with a router conditioned on a learned task embedding. Empirically, the paper claims consistent gains on LIBERO multi-suite training and modest improvements on RoboTwin2 sim/real transfer, while arguing that residualized, dynamically routed RL updates mitigate gradient conflicts and preserve shared features.

**Compliance With Llm Reviewing Policy:**

Affirmed.

**Final Justification:**

The rebuttal has addressed my concerns. I'll maintain my positive score.

**Key Questions For Authors:**

- What is $T_\psi(o,z)$ conditioning on? The observation o includes images, language, proprioception, does the MI critic ingest raw tokens, pooled embeddings, or fused latents? How is o represented inside $T_\psi$?
- If the router is trained with prototypes for a fixed set of N tasks, what mechanism lets it reliably route “previously unseen tasks”? What distribution shift assumptions are required?

**Limitations:**

yes

**Strengths And Weaknesses:**

pros

- The paper focuses on an important topic of robotics.
- The paper is well written and easy to follow.

cons

- The “IB” part empirically seems weak: removing IB drops avg success only from 97.1 to 96.8 (Table 5), a marginal effect relative to the overall gains being attributed to the method.
- There’s a conceptual mismatch between the stated motivation (“discard nuisance factors like background/lighting”) and the actual implementation details: the paper seems do not show any evidence that the representation is actually more invariant or compact.
- The paper claims task embeddings do not rely on explicit labels like task IDs, yet the contrastive loss explicitly uses task identity $\zeta$ with learnable prototypes $e_\zeta = Emb(\zeta)$, and Algorithm 1 samples a task identity each episode. If the approach depends on known task segmentation/IDs during training, then its relevance to “open-world” or unlabeled multi-task streams is significantly weaker than claimed.
- The real-world results are few (4 tasks, 20 trials). The average improvement over the RFT baseline is modest (57.5 vs 55.0), with some tasks tied.

---

> ### Author Rebuttal · Authors · 2026-03-30
>
> Thanks for your in-depth review, positive assessment of our work (e.g., ***important problem setting, clear presentation, easy-to-follow exposition***), and constructive suggestions. Below, we respond to each point in detail.
>
> **W1. The empirical contribution of IB appears limited.**
>
> Thanks for your concern. We would like to clarify that the role of **IB** in our framework is primarily as a **regularizer that makes the learned representation more stable to irrelevant changes**, rather than the main source of the overall performance gain.
>
> To clarify its role, we add OOD evaluation on **LIBERO-Plus [1]** under nuisance-related shifts (**lighting, background, sensor noise**), comparing **DyGRO-VLA** and **DyGRO-VLA w/o IB**.
>
> While IB brings only a small ID gain (**97.1 vs. 96.8**), it consistently improves OOD robustness, raising the OOD average from **74.5** to **77.1**. We therefore position IB as a **supporting robustness regularizer**.
>
> | Method | ID Avg. | Light | Background | Noise | OOD Avg. |
> |---|---:|---:|---:|---:|---:|
> | DyGRO-VLA w/o IB | 96.8 | 85.0 | 84.3 | 54.2 | 74.5 |
> | DyGRO-VLA | **97.1** | **87.3** | **86.9** | **57.0** | **77.1** |
>
> We have revised the paper accordingly to better clarify the role of IB in the overall method.
>
> **W2. The paper does not provide direct evidence that the representation is more invariant or compact**
>
> Thanks for raising this point. Here, **“more invariant”** means the learned representation is **less sensitive to action-irrelevant visual factors** (e.g., lighting, background, sensor noise), while preserving action-relevant cues.
>
> We therefore add a representation-level analysis: we keep the simulator state fixed, apply nuisance-only perturbations (**lighting, background, sensor noise**), and measure the **mean cosine drift** of the learned latent (**lower is better**). We compare **DyGRO-VLA** with the variant **without IB**. DyGRO-VLA consistently shows smaller latent drift, indicating stronger invariance to irrelevant visual changes.
>
> | Method | Light | Background | Noise |
> |---|---:|---:|---:|
> | DyGRO-VLA w/o IB | 0.0112 | 0.0214 | 0.0394 |
> | DyGRO-VLA | **0.0108** | **0.0109** | **0.0257** |
>
> We also clarify that the **IB objective here is not “traditional compression” in a blind sense**. It performs **task-directed compression**, encouraging the latent representation $Z$ to suppress nuisance factors while preserving information useful for predicting the robot action $A$, i.e., favoring higher $I(Z;A)$. We have revised the paper accordingly.
>
> **W3/Q2. Claims about label-free routing and unseen-task generalization**
>
> Thanks for your concern. Label-free routing is used primarily at **inference time**, where routing decisions are conditioned on the learned task embedding rather than a manually provided task-ID token. However, training still requires task labels to structure the task-embedding space through contrastive prototype learning.
>
> Routing for arbitrary unseen tasks or unlabeled open-world task streams remains challenging in our current design. Since the router does not introduce new prototypes at test time, generalization to unseen tasks is mainly plausible under mild-to-moderate compositional shift, where new tasks lie close to the learned task manifold.
>
> We have revised the wording accordingly and removed overly broad claims such as “instead of relying on explicit environment labels” and “including previously unseen ones.” We now describe the method more precisely as **task-ID-free routing at inference**, rather than open-world unlabeled generalization.
>
> **W4. Real-world evidence is limited and gains are modest**
>
> Thanks for raising this concern.Our real-world experiments primarily serve as a proof-of-concept Sim2Real validation, demonstrating competitive transfer to real hardware. The paper’s main contribution, however, lies in improving cross-task scaling and representation preservation during multi-task post-training.
>
> **Q1. What exactly is the observation representation in the IB objective?**
>
> Thank you for pointing this out. In Eq. (3), $o$ serves as a conceptual shorthand for the full observation tuple, including visual inputs, language instruction, and proprioception. In the actual implementation, however, $T_\psi$ does not ingest raw observations directly. Instead, it operates on continuous embedding summaries: the observation-side input is the mean-pooled multimodal embedding sequence before the action head, while the latent-side input is the pooled fused actor latent.
>
> Thus, the implemented IB objective should be understood as an **embedding-level surrogate** for $I(O; Z)$, rather than mutual information computed directly between raw observations $O$ and the latent variable $Z$. We have revised the text to make this distinction explicit.
>
> **References:**
>
> [1] Fei S, Wang S, Shi J, et al. LIBERO-Plus: In-depth robustness analysis of vision-language-action models.

---

> > ### Author Rebuttal · Reviewer_dNXG · 2026-04-03
> >
> > Thanks for the detailed response. I'll maintain my positive score.

---

> > > ### Author Response · Authors · 2026-04-03
> > >
> > > Thank you for your time and expertise during the review. We appreciate your decision to maintain the positive assessment. If you have any further questions, feel free to discuss with us.

---

### Official Review · Reviewer_dbXk · 2026-03-23

**Soundness:** 2
**Presentation:** 2
**Significance:** 2
**Originality:** 3
**Overall Recommendation:** 3
**Confidence:** 4

**Summary:**

This paper studies the challenge of scaling VLA models to many tasks, identifying that RL fine-tuning often harms cross-task generalization by distorting shared representations and causing task interference. To address this, the authors propose DyGRO-VLA, a two-stage framework that first learns a task-sharing latent representation using an Information Bottleneck objective, and then applies RL through a Mixture-of-RL-Residuals , where multiple residual policies specialize in subsets of tasks and are dynamically combined via a learned router. This design preserves a stable base policy while enabling task-specific improvements. Experiments on simulation manipulation benchmarks and real-world settings show that DyGRO-VLA outperforms both offline baselines and standard RL fine-tuning, achieving better multi-task performance and scalability.

**Compliance With Llm Reviewing Policy:**

Affirmed.

**Final Justification:**

My concerns about the limited gains remain unresolved. The evaluation is limited in scope, lacking results on stronger backbones and baseline comparisons for OOD robustness, making it difficult to fully assess the method’s effectiveness. I will keep my score.

**Key Questions For Authors:**

1. Which VLA model is used in Figure 1? The figure does not clearly specify the underlying model.
2. The paper does not clearly describe how the latent variable $\hat{z}_T$ is computed from the fused representation $h_t^{\text{fuse}}$. Could the authors provide more details on this transformation?

Additional questions are discussed in the Weaknesses section.

**Limitations:**

Yes

**Strengths And Weaknesses:**

Strengths:
- The paper studies multi-task generalization in VLA models, which is a fundamental and important challenge in embodied AI. The motivation is clear and relevant to real-world robot learning.
- The use of the IB objective to learn a task-sharing latent representation is well-motivated and grounded in theory. The paper provides some level of theoretical justification for this design.

Weaknesses:
- The model applies an additional fusion module on top of the Qwen2.5 backbone, but it is unclear why this is necessary, given that LLMs already perform multimodal integration via attention.
- Avoiding catastrophic forgetting is a primary motivation, yet the paper does not include explicit experiments measuring forgetting.
- On RoboTwin2 real-world tasks, DyGRO-VLA only slightly outperforms OpenVLA-OFT after RFT fine-tuning, despite introducing additional complexity such as mixture-of-experts and routing.
- The paper does not include stronger supervised fine-tuning baselines such as Pi0.5[1], which are known to outperform pi0-FAST.

[1] Intelligence, Physical, et al. ": a Vision-Language-Action Model with Open-World Generalization." arXiv preprint arXiv:2504.16054 (2025).

---

> ### Author Rebuttal · Authors · 2026-03-30
>
> Thanks for your in-depth review, positive assessment of our work (e.g., ***important problem setting, clear motivation, principled design***), and constructive suggestions. Below, we respond to each point in detail.
>
> **W1. Why is an additional fusion module needed on top of the Qwen2.5 backbone?**
>
> Thanks for raising this important point. We do not claim that Qwen2.5 cannot perform multimodal integration by itself. Rather, we introduce the fusion module because the backbone provides general-purpose multimodal features, whereas robot control requires a more ***structured, action-oriented latent***. Specifically, the fusion module provides three benefits beyond backbone attention: (1) it ***injects proprioceptive signals***, which are critical for manipulation but not natively represented in the language backbone; (2) it ***maps conditioning tokens, action tokens, and proprioceptive input into an action-chunk-aligned control latent*** through learnable action queries; and (3) it enables this control-specific adaptation in a lightweight module, helping ***preserve the backbone’s shared representation*** during training.
>
> Therefore, the fusion module does not duplicate the backbone’s multimodal reasoning; it converts general multimodal features into proprio-conditioned control latents for the action head and residual router. We have clarified this motivation in the revision.
>
> **W2. Catastrophic forgetting is a key motivation, but there is no explicit forgetting experiment.**
>
> Thanks for raising this point. Fig. 1, Fig. 2, and Fig. 3 already show that standard RFT suffers from cross-suite degradation, task-scaling instability, and representation drift. To clarify this better, we have added a direct comparison of **train-suite and cross-suite performance over post-training steps** for standard RFT versus DyGRO-VLA, where both methods are post-trained on LIBERO-Spatial and evaluated on LIBERO-Spatial and LIBERO-Object. The corresponding figure is provided as Figure A at our anonymous website (https://anonymous.4open.science/w/DyGRO/).
>
> **W3. Limited Performance on RoboTwin2 real-world tasks.**
>
> Thanks for raising this concern. Our main contribution is not broad real-world gain, but improved **cross-task post-training scalability** under RL finetuning. In this paper, RoboTwin2 serves primarily as a **proof-of-concept Sim2Real validation** showing that the learned policy can transfer competitively to real hardware. We have revised the discussion to make this positioning clearer. Since the compared methods use different pretrained foundations and model scales, we view this result primarily as evidence of competitive transferability rather than a definitive comparison of absolute real-world performance.
>
> **W4. The paper does not include stronger supervised finetuning baselines such as Pi0.5.**
>
> Thanks for raising this concern. We have added pi0 and pi0.5 results to the LIBERO comparison table:
>
> **Table R2. Additional SFT baselines on LIBERO**
> | Model | Spatial | Object | Goal | Long | Avg |
> |---|---:|---:|---:|---:|---:|
> | pi0.5 | **98.8** | 98.2 | **98.0** | 92.4 | 96.9 |
> | pi0 | 98.0 | 96.8 | 94.4 | 88.4 | 94.4 |
> | DyGRO-VLA | 97.6 | **98.6** | 97.2 | **95.0** | **97.1** |
>
> **Q1. Which VLA model is used in Figure 1?**
>
> Thank you for pointing this out. Figure 1 uses the base VLA model defined in Sec. 4.1, rather than the full DyGRO model with MoRR. We agree that the original caption was not sufficiently explicit, and we have revised the figure caption and surrounding text to clarify this point.
>
> **Q2. How is the latent variable $\tilde{z}^T$ computed from the fused representation $h_t^{\text{fuse}}$ ?**
>
> Thanks for this point. Given the fused representation $h_t^{\text{fuse}}$, we first apply an attention-pooling operator to obtain a global task-conditioned feature, and then use a lightweight projection head to produce the latent variable:
>
> $$
> g_t = \mathrm{AttnPool}(h_t^{\text{fuse}}), \qquad
> \tilde{z}^T = \mathrm{MLP}(g_t).
> $$
>
> This latent is then used as the task feature for downstream policy prediction. We have added this implementation detail to the method section to clarify how the latent variable is computed.

---

> > ### Author Rebuttal · Reviewer_dbXk · 2026-04-04
> >
> > Thank you for your detailed rebuttal. My concerns regarding the motivation and the clarification of the method design have been addressed. However, the limited gains in both the simulation (compared to pi0.5) and real-world experiments remain my main concern. I will maintain my current score.

---

> > > ### Author Response · Authors · 2026-04-07
> > >
> > > We thank the reviewer for the thoughtful follow-up and for acknowledging that our clarification on the motivation and method design has addressed the earlier concerns. Regarding the remaining concerns about the **limited gains**, we address them from the following perspectives.
> > >
> > > **1. The main contribution of DyGRO**
> > >
> > > We want to emphasize that our claim is not that DyGRO universally outperforms much larger robotics-pretrained VLAs in absolute performance. Rather, the goal of this paper is to improve **cross-task RL post-training scalability** by preserving shared task representations during online adaptation. In this sense, the key question is whether DyGRO improves the **same lightweight base VLA** under RL post-training. Across our ablations and comparisons, the answer is consistently yes: DyGRO improves the base model in cross-task post-training performance, reduces degradation across suites, and yields more stable task scaling under RL finetuning.
> > >
> > > To further examine whether the benefit of DyGRO-VLA goes beyond standard in-distribution average success rate, we additionally evaluate it on **LIBERO-Plus [1]** under nuisance-only distribution shifts. In this setting, we keep the underlying simulator state unchanged and introduce test-time perturbations, including **lighting**, **background**, and **sensor noise**. As shown in **Table A**, under nuisance shifts, **DyGRO-VLA achieves the highest OOD average success rate (77.1%) and the smallest ID-to-OOD drop (20.0 points)** among the compared variants. We view this as supporting our main claim that DyGRO improves the robustness and transferability of shared representations during cross-task RL post-training.
> > >
> > > **Table A. OOD robustness on LIBERO-Plus.**
> > > | Method | ID Avg (Succ. ↑) | Light (Succ. ↑) | Background (Succ. ↑) | Noise (Succ. ↑) | OOD Avg (Succ. ↑) |
> > > |---|---:|---:|---:|---:|---:|
> > > | DyGRO-VLA (SFT) | 92.6 | 80.4 | 77.2 | 48.6 | 68.7 |
> > > | DyGRO-VLA w/o IB | 96.8 | 85.0 | 84.3 | 54.2 | 74.5 |
> > > | DyGRO-VLA | 97.1 | 87.3 | 86.9 | 57.0 | 77.1 |
> > >
> > > **2. Model scale and pretraining matter**
> > >
> > > In comparing the performance with **pi0.5**, we believe the modest absolute margin is related to the substantial gap in **model size** and **robotics pretraining**. **DyGRO-VLA** is built upon a **Qwen2.5-0.5B** backbone and does **not** rely on a robotics-pretrained VLA base, whereas **pi0.5** utilizes a substantially larger model with extensive robotics pretraining. Similarly, **OpenVLA-OFT** is based on a significantly larger, robotics-pretrained model, as summarized in **Table B**. Despite this clear disparity in both model capacity and robotics pretraining, **DyGRO-VLA** achieves performance that is already competitive with **pi0.5**, suggesting that the proposed post-training strategy is genuinely effective, rather than merely benefiting from a stronger pretrained foundation.
> > >
> > > **Table B. Backbone scale and robotics pretraining of compared models.**
> > > | Model | Backbone Scale | Pretraining |
> > > |---|---|---|
> > > | **DyGRO-VLA** | 0.5B | No robotics-pretrained VLA base |
> > > | **pi0.5** | 2B | Heterogeneous robot + web + high-level co-training |
> > > | **OpenVLA-OFT** | 7B | 970k real-world robot demonstrations |
> > >
> > > We have revised the paper to better explain this point and to clarify why, given the substantial gap in model scale and robotics pretraining, achieving competitive performance with a lightweight 0.5B backbone is already meaningful. We thank the reviewer again for this thoughtful comment and hope that the clarification above helps address the remaining concerns.
> > >
> > > [1] Fei et al. *LIBERO-Plus: In-depth robustness analysis of vision-language-action models*.

---

### Decision · Program_Chairs · 2026-04-30

**Decision:**

Accept (regular)

**Comment:**

This paper proposes a novel algorithm using RL to finetune VLAs without sacrificing generalizability. The reviewers generally agreed that the paper is tackling an important problem with an interesting approach. There were concerns about the experiments, including missing baselines, missing analyses, and the scope of the real-world experiments. Several of these concerns were addressed during the rebuttal period, though there were some lingering concerns about the comparison to pi0.5 and to SFT baselines. Addressing these concerns can improve the quality of the paper.